# Robust Learning with Progressive Data Expansion Against Spurious Correlation

**Yihe Deng**[*]  **Yu Yang**[*]  **Baharan Mirzasoleiman**  **Quanquan Gu**
Department of Computer Science
University of California, Los Angeles
Los Angeles, CA 90095
{yihedeng,yuyang,baharan,qgu}@cs.ucla.edu

## Abstract

While deep learning models have shown remarkable performance in various tasks, they are susceptible to learning non-generalizable *spurious features* rather than the core features that are genuinely correlated to the true label. In this paper, beyond existing analyses of linear models, we theoretically examine the learning process of a two-layer nonlinear convolutional neural network in the presence of spurious features. Our analysis suggests that imbalanced data groups and easily learnable spurious features can lead to the dominance of spurious features during the learning process. In light of this, we propose a new training algorithm called **PDE** that efficiently enhances the model's robustness for a better worst-group performance. PDE begins with a group-balanced subset of training data and progressively expands it to facilitate the learning of the core features. Experiments on synthetic and real-world benchmark datasets confirm the superior performance of our method on models such as ResNets and Transformers. On average, our method achieves a $2.8\%$ improvement in worst-group accuracy compared with the state-of-the-art method, while enjoying up to $10\times$ faster training efficiency. Codes are available at `https://github.com/uclaml/PDE`.

## 1 Introduction

Despite the remarkable performance of deep learning models, recent studies (Sagawa et al., 2019, 2020; Izmailov et al., 2022; Haghtalab et al., 2022; Yang et al., 2022, 2023b,c) have identified their vulnerability to spurious correlations in data distributions. A spurious correlation refers to an easily learned feature that, while unrelated to the task at hand, appears with high frequency within a specific class. For instance, waterbirds frequently appear with water backgrounds, and landbirds with land backgrounds. When training with empirical risk minimization (ERM), deep learning models tend to exploit such correlations and fail to learn the more subtle features genuinely correlated with the true labels, resulting in poor generalization performance on minority data (e.g., waterbirds with land backgrounds as shown in Figure 1). This observation raises a crucial question: *Does the model genuinely learn to classify birds, or does it merely learn to distinguish land from water?* The issue is particularly concerning because deep learning models are being deployed in critical applications such as healthcare, finance, and autonomous vehicles, where we require a reliable predictor.

Researchers formalized the problem by considering examples with various combinations of core features (e.g., landbird/waterbird) and spurious features (e.g., land/water backgrounds) as different *groups*. The model is more likely to make mistakes on certain groups if it learns the spurious feature. The objective therefore becomes balancing and improving performance across all groups. Under this formulation, we can divide the task into two sub-problems: (1) accurately identifying the groups, which are not always known in a dataset, and (2) effectively using the group information

---

[*]Equal contribution

37th Conference on Neural Information Processing Systems (NeurIPS 2023).

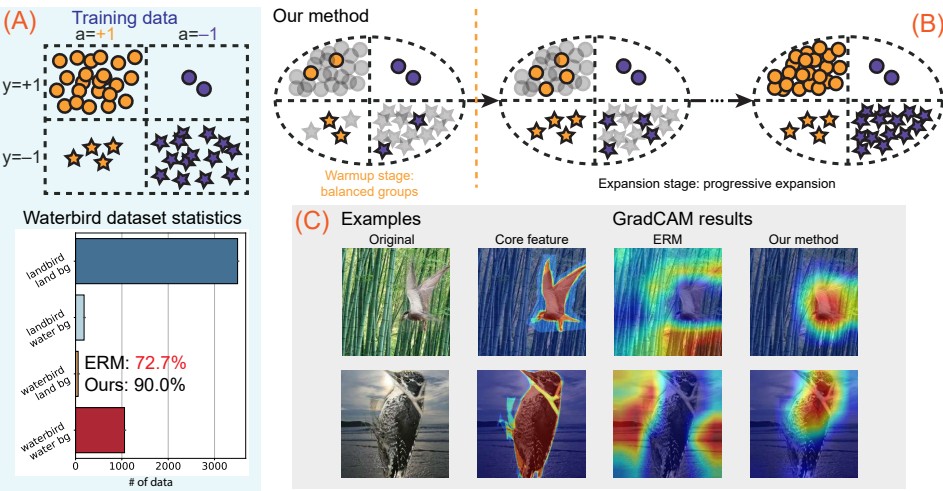

Figure 1: A overview of the problem, our proposed solution, and the resultant outcomes. (A) We demonstrate the data distribution and provide an example of the statistics of Waterbirds. (B) The overall procedure of PDE. (C) we use GradCAM (Selvaraju et al., 2017) to show the attention of the model trained with PDE as compared to ERM. While ERM focuses on the background, PDE successfully trains the model to capture the birds.

to finally improve the model's robustness. While numerous recent works (Nam et al., 2020; Liu et al., 2021; Creager et al., 2021; Ahmed et al., 2021; Taghanaki et al., 2021; Zhang et al., 2022) focus on the first sub-problem, the second sub-problem remains under studied. The pioneering work (Sagawa et al., 2019) still serves as the best guidance for utilizing accurate group information. In this paper, we focus on the second sub-problem and aim to provide a more effective and efficient algorithm to utilize the group information. It is worth noting that the theoretical understanding of spurious correlations lags behind the empirical advancements of mitigating spurious features. Existing theoretical studies (Sagawa et al., 2020; Chen et al., 2020; Yang et al., 2022; Ye et al., 2022) are limited to the setting of simple linear models and data distribution that are less reflective of real application scenarios.

We begin by theoretically examining the learning process of spurious features when training a two-layer nonlinear convolutional neural network (CNN) on a corresponding data model that captures the influence of spurious correlations. We illustrate that the learning of spurious features swiftly overshadows the learning of core features from the onset of training when groups are imbalanced and spurious features are more easily learned than core features. Based upon our theoretical understanding, we propose Progressive Data Expansion (**PDE**), a neat and novel training algorithm that efficiently uses group information to enhance model's robustness against spurious correlations. Existing approaches, such as GroupDRO (Sagawa et al., 2019) and upsampling techniques (Liu et al., 2021), aim to balance the data groups in each batch throughout the training process. In contrast, we employ a small balanced warm-up subset only at the beginning of the training. Following a brief period of balanced training, we progressively expand the warm-up subset by adding small random subsets of the remaining training data until using all of them, as shown in the top right of Figure 1. Here, we utilize the momentum from the warm-up subset to prevent the model from learning spurious features when adding new data. Empirical evaluations on both synthetic and real-world benchmark data validate our theoretical findings and confirm the effectiveness of PDE. Additional ablation studies also demonstrate the significance and impact of each component within our training scheme. In summary, our contributions are highlighted as follows:

- We provide a theoretical understanding of the impact of spurious correlations beyond the linear setting by considering a two-layer nonlinear CNN.
- We introduce PDE, a theory-inspired approach that effectively addresses the challenge posed by spurious correlations.
  - PDE achieves the best performance on benchmark vision and language datasets for models including ResNet and Transformer. On average, it outperforms the state-of-the-art method by 2.8% in terms of worst-group accuracy.
  - PDE enjoys superior training efficiency, being $10\times$ faster than the state-of-the-art methods.

## 2 Why is Spurious Correlation Harmful to ERM?

In this section, we simplify the intricate real-world problem of spurious correlations into a theoretical framework. We provide analysis on two-layer nonlinear CNNs, extending beyond the linear setting prevalent in existing literature on this subject. Under this framework, we formally present our theory concerning the training process of empirical risk minimization (ERM) in the presence of spurious features. These theoretical insights motivate the design of our algorithm.

### 2.1 Empirical Risk Minimization

We begin with the formal definition of the ERM-based training objective for a binary classification problem. Consider a training dataset $S = \{(\mathbf{x}_i, y_i)\}_{i=1}^N$, where $\mathbf{x}_i \in \mathbb{R}^d$ is the input and $y \in \{\pm 1\}$ is the output label. We train a model $f(\mathbf{x}; \mathbf{W})$ with weight $\mathbf{W}$ to minimize the empirical loss function:

$$\mathcal{L}(\mathbf{W}) = \frac{1}{N}\sum_{i=1}^N \ell\big(y_i f(\mathbf{x}_i; \mathbf{W})\big), \tag{2.1}$$

where $\ell$ is the logistic loss defined as $\ell(z) = \log(1 + \exp(-z))$. The empirical risk minimizer refers to $\mathbf{W}^*$ that minimizes the empirical loss: $\mathbf{W}^* := \arg\min_{\mathbf{W}} \mathcal{L}(\mathbf{W})$. Typically, gradient-based optimization algorithms are employed for ERM. For example, at each iteration $t$, gradient descent (GD) has the following update rule:

$$\mathbf{W}^{(t+1)} = \mathbf{W}^{(t)} - \eta \nabla \mathcal{L}(\mathbf{W}^{(t)}). \tag{2.2}$$

Here, $\eta > 0$ is the learning rate. In the next subsection, we will show that even for a relatively simple data model which consists of core features and spurious features, vanilla ERM will fail to learn the core features that are correlated to the true label.

### 2.2 Data Distribution with Spurious Correlation Fails ERM

Previous work such as (Sagawa et al., 2020) considers a data model where the input consists of core feature, spurious feature and noise patches at fixed positions, i.e., $\mathbf{x} = [\mathbf{x}_{\text{core}}, \mathbf{x}_{\text{spu}}, \mathbf{x}_{\text{noise}}]$. In real-world applications, however, features in an image do not always appear at the same pixels. Hence, we consider a more realistic data model where the patches do not appear at fixed positions.

**Definition 2.1** (Data model). A data point $(\mathbf{x}, y, a) \in (\mathbb{R}^d)^P \times \{\pm 1\} \times \{\pm 1\}$ is generated from the distribution $\mathcal{D}$ as follows.

- Randomly generate the true label $y \in \{\pm 1\}$.
- Generate spurious label $a \in \{\pm y\}$, where $a = y$ with probability $\alpha > 0.5$.
- Generate $\mathbf{x}$ as a collection of $P$ patches: $\mathbf{x} = (\mathbf{x}^{(1)}, \mathbf{x}^{(2)}, \dots, \mathbf{x}^{(P)}) \in (\mathbb{R}^d)^P$, where
  - **Core feature.** One and only one patch is given by $\beta_c \cdot y \cdot \mathbf{v}_c$ with $\|\mathbf{v}_c\|_2 = 1$.
  - **Spurious feature.** One and only one patch is given by $\beta_s \cdot a \cdot \mathbf{v}_s$ with $\|\mathbf{v}_s\|_2 = 1$ and $\langle \mathbf{v}_c, \mathbf{v}_s \rangle = 0$.
  - **Random noise.** The rest of the $P - 2$ patches are Gaussian noises $\boldsymbol{\xi}$ that are independently drawn from $N(0, (\sigma_p^2/d) \cdot \mathbf{I}_d)$ with $\sigma_p$ as an absolute constant.

  And $0 < \beta_c \ll \beta_s \in \mathbb{R}$.

Similar data models have also been considered in recent works on feature learning (Allen-Zhu & Li, 2020; Zou et al., 2021; Chen et al., 2022; Jelassi & Li, 2022), where the input data is partitioned into feature and noise patches. We extend their data models by further positing that certain feature patches might be associated with the spurious label instead of the true label. In the rest of the paper, we assume $P = 3$ for simplicity. With the given data model, we consider the training dataset $S = \{(\mathbf{x}_i, y_i, a_i)\}_{i=1}^N$ and let $S$ be partitioned into large group $S_1$ and small group $S_2$ such that $S_1$ contains all the training

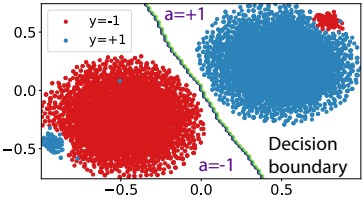

Figure 2: Visualization of the data.

data that can be correctly classified by the spurious feature, i.e., $a_i = y_i$, and $S_2$ contains all the training data that can only be correctly classified by the core feature, i.e., $a_i = -y_i$. We denote $\widehat{\alpha} = \frac{|S_1|}{N}$ and therefore $1 - \widehat{\alpha} = \frac{|S_2|}{N}$.

**Visualization of our data.** In Figure 2, we present the visualization in 2D space of the higher-dimensional data generated from our data model using t-SNE (Van der Maaten & Hinton, 2008), where data within each class naturally segregate into large and small groups. The spurious feature is sufficient for accurate classification of the larger group data, but will lead to misclassification of the small group data.

## 2.3 Beyond Linear Models

We consider a two-layer nonlinear CNN defined as follows:

$$f(\mathbf{x}; \mathbf{W}) = \sum_{j \in [J]} \sum_{p=1}^{P} \sigma(\langle \mathbf{w}_j, \mathbf{x}^{(p)} \rangle), \tag{2.3}$$

where $\mathbf{w}_j \in \mathbb{R}^d$ is the weight vector of the $j$-th filter, $J$ is the number of filters (neurons) of the network, and $\sigma(z) = z^3$ is the activation function. $\mathbf{W} = [\mathbf{w}_1, \dots, \mathbf{w}_J] \in \mathbb{R}^{d \times J}$ denotes the weight matrix of the CNN. Similar two-layer CNN architectures are analyzed in in (Chen et al., 2022; Jelassi & Li, 2022) but for different problems, where the cubic activation serves a simple function that provides non-linearity. Similar to Jelassi & Li (2022); Cao et al. (2022), we assume a mild overparameterization of the CNN with $J = \text{polylog}(d)$. We initialize $\mathbf{W}^{(0)} \sim \mathcal{N}(0, \sigma_0^2)$, where $\sigma_0^2 = \frac{\text{polylog}(d)}{d}$. Due to the CNN structure, our analysis can handle data models where each data can have arbitrary order of patches while linear models fail to do so.

## 2.4 Understanding the Training Process with Spurious Correlation

In this subsection, we formally introduce our theoretical result on the training process of the two-layer CNN using gradient descent in the presence of spurious features. We first define the performance metrics. A frequently considered metric is the test accuracy: $\text{Acc}(\mathbf{W}) = \mathbb{P}_{(\mathbf{x}, y, a) \sim \mathcal{D}}[\text{sgn}(f(\mathbf{x}; \mathbf{W})) = y]$. With spurious correlations, researchers are more interested in the worst-group accuracy:

$$\text{Acc}_{\text{wg}}(\mathbf{W}) = \min_{y \in \{\pm 1\}, a \in \{\pm 1\}} \mathbb{P}_{(\mathbf{x}, y, a) \sim \mathcal{D}}[\text{sgn}(f(\mathbf{x}; \mathbf{W})) = y],$$

which accesses the worst accuracy of a model among all groups defined by combinations of $y$ and $a$. We then summarize the learning process of ERM in the following theorem. Our analysis focuses on the learning of spurious and core features, represented by the growth of $\langle \mathbf{w}_i^{(t)}, \mathbf{v}_s \rangle$ and $\langle \mathbf{w}_i^{(t)}, \mathbf{v}_c \rangle$ respectively:

**Theorem 2.2.** Consider the training dataset $S = \{(\mathbf{x}_i, y_i)\}_{i=1}^{N}$ that follows the distribution in Definition 2.1. Consider the two-layer nonlinear CNN model as in (2.3) initialized with $\mathbf{W}^{(0)} \sim \mathcal{N}(0, \sigma_0^2)$. After training with GD in (2.2) for $T_0 = \widetilde{\Theta}(1/(\eta \beta_s^3 \sigma_0))$ iterations, for all $j \in [J]$ and $t \in [0, T_0)$, we have

$$\widetilde{\Theta}(\eta) \beta_s^3 (2\widehat{\alpha} - 1) \cdot \langle \mathbf{w}_j^{(t)}, \mathbf{v}_s \rangle^2 \leq \langle \mathbf{w}_j^{(t+1)}, \mathbf{v}_s \rangle - \langle \mathbf{w}_j^{(t)}, \mathbf{v}_s \rangle \leq \widetilde{\Theta}(\eta) \beta_s^3 \widehat{\alpha} \cdot \langle \mathbf{w}_j^{(t)}, \mathbf{v}_s \rangle^2, \tag{2.4}$$

$$\widetilde{\Theta}(\eta) \beta_c^3 \widehat{\alpha} \cdot \langle \mathbf{w}_j^{(t)}, \mathbf{v}_c \rangle^2 \leq \langle \mathbf{w}_j^{(t+1)}, \mathbf{v}_c \rangle - \langle \mathbf{w}_j^{(t)}, \mathbf{v}_c \rangle \leq \widetilde{\Theta}(\eta) \beta_c^3 \cdot \langle \mathbf{w}_j^{(t)}, \mathbf{v}_c \rangle^2. \tag{2.5}$$

After training for $T_0$ iterations, with high probability, the learned weight has the following properties: (1) it learns the spurious feature $\mathbf{v}_s$: $\max_{j \in [J]} \langle \mathbf{w}_j^{(T)}, \mathbf{v}_s \rangle \geq \widetilde{\Omega}(1/\beta_s)$; (2) it *almost* does not learn the core feature $\mathbf{v}_c$: $\max_{j \in [J]} \langle \mathbf{w}_j^{(T)}, \mathbf{v}_c \rangle = \widetilde{\mathcal{O}}(\sigma_0)$.

**Discussion.** The detailed proof is deferred to Appendix E, and we provide intuitive explanations of the theorem as follows. A larger value of $\langle \mathbf{w}_i^{(t)}, \mathbf{v} \rangle$ for $\mathbf{v} \in \{\mathbf{v}_s, \mathbf{v}_c\}$ implies better learning of the feature vector $\mathbf{v}$ by neuron $\mathbf{w}_i$ at iteration $t$. As illustrated in (2.4) and (2.5), the updates for both spurious and core features are non-zero, as they depend on the squared terms of themselves with non-zero coefficients, while the growth rate of $\langle \mathbf{w}_i^{(t)}, \mathbf{v}_s \rangle$ is significantly faster than that of $\langle \mathbf{w}_i^{(t)}, \mathbf{v}_c \rangle$. Consequently, the neural network rapidly learns the spurious feature but barely learns the core feature, as it remains almost unchanged from initialization.

We derive the neural network's prediction after training for $T_0$ iterations. For a randomly generated data example $(\mathbf{x}, y, a) \sim \mathcal{D}$, the neural network's prediction is given by $\text{sgn}(f(\mathbf{x}; \mathbf{W})) = \text{sgn}(\sum_{j \in [J]} (y\beta_c^3 \langle \mathbf{w}_j, \mathbf{v}_c \rangle^3 + a\beta_s^3 \langle \mathbf{w}_j, \mathbf{v}_s \rangle^3 + \langle \mathbf{w}_j, \xi \rangle^3))$. Since the term $\beta_s^3 \max_{j \in [J]} \langle \mathbf{w}_j, \mathbf{v}_s \rangle^3$ dominates the summation, the prediction will be $\text{sgn}(f(\mathbf{x}; \mathbf{W})) = a$. Consequently, we obtain the test accuracy as $\text{Acc}(\mathbf{W}) = \alpha$, since $a = y$ with probability $\alpha$, and the model accurately classifies the large group. However, when considering the small group and examining examples where $y \neq a$, the models consistently make errors, resulting in $\text{Acc}_{wg}(\mathbf{W}) = 0$. To circumvent this poor performance on worst-group accuracy, an algorithm that can avoid learning the spurious feature is in demand.

## 3 Theory-Inspired Two-Stage Training Algorithm

In this section, we introduce Progressive Data Expansion (PDE), a novel two-stage training algorithm inspired by our analysis to enhance robustness against spurious correlations. We begin with illustrating

the implications of our theory, where we provide insights into the data distributions that lead to the rapid learning of spurious features and clarify scenarios under which the model remains unaffected.

## 3.1 Theoretical Implications

Notably in Theorem 2.2, the growth of the two sequences $\langle \mathbf{w}_i^{(t)}, \mathbf{v}_s \rangle$ in (2.4) and $\langle \mathbf{w}_i^{(t)}, \mathbf{v}_c \rangle$ in (2.5) follows the formula $x_{t+1} = x_t + \eta A x_t^2$, where $x_t$ represents the inner product sequence with regard to iteration $t$ and $A$ is the coefficient containing $\widehat{\alpha}$, $\beta_c$ or $\beta_s$. This formula is closely related to the analysis of tensor power methods (Allen-Zhu & Li, 2020). In simple terms, when two sequences have slightly different growth rates, one of them will experience much faster growth in later times. As we will show below, the key factors that determine the drastic difference of spurious and core features in later times are the group size $\widehat{\alpha}$ and feature strengths $\beta_c, \beta_s$.

- **When the model learns spurious feature ($\beta_c^3 < \beta_s^3 (2\widehat{\alpha} - 1)$).** We examine the lower bound for the growth of $\langle \mathbf{w}_i^{(t)}, \mathbf{v}_s \rangle$ in (2.4) and the upper bound for the growth of $\langle \mathbf{w}_i^{(t)}, \mathbf{v}_c \rangle$ in (2.5) in Theorem 2.2. If $\beta_c^3 < \beta_s^3 (2\widehat{\alpha} - 1)$, we can employ the tensor power method and deduce that the spurious feature will be learned first and rapidly. The condition on data distribution imposes two necessary conditions: $\widehat{\alpha} > 1/2$ (groups are imbalanced) and $\beta_c < \beta_s$ (the spurious feature is stronger). This observation is consistent with real-world datasets, such as the Waterbirds dataset, where $\widehat{\alpha} = 0.95$ and the background is much easier to learn than the intricate features of the birds.
- **When the model learns core feature ($\beta_c > \beta_s$).** However, if we deviate from the aforementioned conditions and consider $\beta_c > \beta_s$, we can examine the lower bound for the growth of $\langle \mathbf{w}_i^{(t)}, \mathbf{v}_c \rangle$ in (2.5) and the upper bound for the growth of $\langle \mathbf{w}_i^{(t)}, \mathbf{v}_s \rangle$ in (2.4). Once again, we apply the tensor power method and determine that the model will learn the core feature rapidly. In real-world datasets, this scenario corresponds to cases where the core feature is not only significant but also easier to learn than the spurious feature. Even for imbalanced groups with $\widehat{\alpha} > 1/2$, the model accurately learns the core feature. Consequently, enhancing the coefficients of the growth of the core feature allows the model to tolerate imbalanced groups. We present verification through synthetic experiments in the next section.

As we will show in the following subsection, we initially break the conditions of learning the spurious feature by letting $\widehat{\alpha} = 1/2$ in a group-balanced data subset. Subsequently, we utilize the momentum to amplify the core feature's coefficient, allowing for tolerance of $\widehat{\alpha} > 1/2$ when adding new data.

## 3.2 PDE: A Two-Stage Training Algorithm

We present a new algorithm named Progressive Data Expansion (PDE) in Algorithm 1 and explain the details below, which consist of (1) warm-up and (2) expansion stages.

---

**Algorithm 1** Progressive Data Expansion (PDE)

---

**Require:** Number of iterations $T_0$ for warm-up training; number of times $K$ for dataset expansion; number of iterations $J$ for expansion training; number of data $m$ for each expansion; learning rate $\eta$; momentum coefficient $\gamma$; initialization scale $\sigma_0$; training set $S = \{(\mathbf{x}_i, y_i, a_i)\}_{i=1}^n$; model $f_\mathbf{W}$.

1: Initialize $\mathbf{W}^{(0)}$.
   **Warm-up stage**
2: Divide the $S$ into groups by values of $y$ and $a$: $S_{y,a} = \{(\mathbf{x}_i, y_i, a_i)\}_{y_i=y, a_i=a}$.
3: Generate warm-up set $S^0$ from $S$ by randomly subsampling from each group of $S$ such that $|S_{y,a}^0| = \min_{y', a'} |S_{y', a'}|$ for $y \in \{\pm 1\}$ and $a \in \{\pm 1\}$.
4: **for** $t = 0, 1, \ldots, T_0$ **do**
5:    Compute loss on $S^0$: $\mathcal{L}_{S^0}(\mathbf{W}^{(t)}) = \frac{1}{|S^0|} \sum_{i \in S^0} \ell(y_i f(x_i; \mathbf{W}^{(t)}))$.
6:    Update $\mathbf{W}^{(t+1)}$ by (3.1) and (3.2).
7: **end for**
   **Expansion stage**
8: **for** $k = 1, \ldots, K$ **do**
9:    Draw $m$ examples ($S_{[m]}$) from $S/S^{k-1}$ and let $S^k = S^{k-1} \cup S_{[m]}$.
10:    **for** $t = 1, \ldots, J$ **do**
11:      Compute loss on $S^k$: $\mathcal{L}_{S^k}(\mathbf{W}^{(T)}) = \frac{1}{|S^k|} \sum_{i \in S^k} \ell(y_i f(x_i; \mathbf{W}^{(T)}))$, where $T = T_0 + (k-1) * J + t$.
12:      Update $\mathbf{W}^{(T+1)}$ by (3.1) and (3.2).
13:    **end for**
14: **end for**
15: **return** $\mathbf{W}^{(t)} = \mathrm{argmax}_{\mathbf{W}^{(t')}} \mathrm{Acc}_{\mathrm{wg}}^{\mathrm{val}}(\mathbf{W}^{(t')})$.

---

As accelerated gradient methods are most commonly used in applications, we jointly consider the property of momentum and our theoretical insights when designing the algorithm. For gradient descent with momentum (GD+M), at each iteration $t$ and with momentum coefficient $\gamma > 0$, it updates as follows

$$\mathbf{g}^{(t+1)} = \gamma \mathbf{g}^{(t)} + (1-\gamma)\nabla\mathcal{L}(\mathbf{W}^{(t)}), \tag{3.1}$$

$$\mathbf{W}^{(t+1)} = \mathbf{W}^{(t)} - \eta \cdot \mathbf{g}^{(t+1)}, \tag{3.2}$$

**Warm-up Stage.** In this stage, we create a fully balanced dataset $S^0$, in which each group is randomly subsampled to match the size of the smallest group, and consider it as a warm-up dataset. We train the model on the warm-up dataset for a fixed number of epochs. During this phase, the model is anticipated to accurately learn the core feature without being influenced by the spurious feature. Note that, under our data model, a completely balanced dataset will have $\widehat{\alpha} = 1/2$. We present the following lemma as a theoretical basis for the warm-up stage.

**Lemma 3.1.** Given the balanced training dataset $S^0 = \{(\mathbf{x}_i, y_i, a_i)\}_{i=1}^{N_0}$ with $\widehat{\alpha} = 1/2$ as in Definition 2.1 and CNN as in (2.3). The gradient on $\mathbf{v}_s$ will be 0 from the beginning of training.

In particular, with $\widehat{\alpha} = 1/2$ we have $|S_1^0| = |S_2^0|$: an equal amount of data is positively correlated with the spurious feature as the data negatively correlated with the spurious feature. In each update, both groups contribute nearly the same amount of spurious feature gradient with different signs, resulting in cancellation. Ultimately, this prevents the model from learning the spurious feature. Detailed proofs can be found in Appendix F.

**Expansion Stage.** In this stage, we proceed to train the model by incrementally incorporating new data into the training dataset. The rationale for this stage is grounded in the theoretical result by the previous work (Jelassi & Li, 2022) on GD with momentum, which demonstrates that once gradient descent with momentum initially increases its correlation with a feature $\mathbf{v}$, it retains a substantial historical gradient in the momentum containing $\mathbf{v}$. Put it briefly, the initial learning phase has a considerable influence on subsequent training for widely-used accelerated training algorithms. While ERM learns the spurious feature $\mathbf{v}_s$ and momentum does not help, as we will show in synthetic experiments, PDE avoids learning $\mathbf{v}_s$ and learns $\mathbf{v}_c$ in the warm-up stage. This momentum from warm-up, in turn, amplifies the core feature that is present in the gradients of newly added data, facilitating the continued learning of $\mathbf{v}_c$ in the expansion stage. For a specific illustration, the learning of the core feature by GD+M will be

$$\langle \mathbf{w}_j^{(t+1)}, \mathbf{v}_c \rangle = \langle \mathbf{w}_j^{(t)} - \eta(\gamma g^{(t)} + (1-\gamma)\nabla_{\mathbf{w}_j}\mathcal{L}(\mathbf{W}^{(t)})), \mathbf{v}_c \rangle,$$

where $g^{(t)}$ is the additional momentum as compared to GD with $\gamma = 0$. While the current gradient along $\mathbf{v}_c$ might be small (i.e., $\beta_c$), we can benefit from the historical gradient in $g^{(t)}$ to amplify the growth of $\langle \mathbf{w}_j^{(t+1)}, \mathbf{v}_c \rangle$ and make it larger than that of the spurious feature (i.e., $\beta_s$). This learning process will then correspond to the case when the model learns the core feature discussed in Subsection 3.1. Practically, we consider randomly selecting $m$ new examples for expansion every $J$ epochs by attempting to draw a similar number of examples from each group. During the last few epochs of the expansion stage, we expect the newly incorporated data exclusively from the larger group, as the smaller groups have been entirely integrated into the warm-up dataset.

It is worth noting that while many works address the issue of identifying groups from datasets containing spurious correlations, we assume the group information is known and our algorithm focuses on the crucial subsequent question of optimizing group information utilization. Aiming to prevent the learning of spurious features, PDE distinguishes itself by employing a rapid and lightweight warm-up stage and ensuring continuous improvement during the expansion stage with the momentum acquired from the warm-up dataset. Our training framework is both concise and effective, resulting in computational efficiency and ease of implementation.

## 4 Experiments

In this section, we present the experiment results from both synthetic and real datasets. Notably, we report the worst-group accuracy, which assesses the minimum accuracy across all groups and is commonly used to evaluate the model's robustness against spurious correlations.

### 4.1 Synthetic Data

In this section, we present synthetic experiment results in verification to our theoretical findings. In Appendix A, we illustrate the detailed data distribution, hyper-parameters of the experiments and more extensive experiment results. The data used in this seciton is generated following Definition 2.1. We consider the worst-group and overall test accuracy. As illustrated in Table 1, ERM, whether trained with GD or GD+M, is unable to accurately predict the small group in our specified data distribution where $\widehat{\alpha} = 0.98$ and $\beta_c < \beta_s$. In contrast, our method significantly improves worst-group accuracy while maintaining

Table 1: Synthetic data experiments. We report the worst-group accuracy and the gap (i.e., overall - worst). We further consider several variations of PDE to demonstrate the importance of each component of our method. **Reset**: we reset the momentum to zero after the warm-up stage. **Warmup+All**: we let PDE to incorporate all of the new training data at once after the warm-up stage.

|  | Worst-group (%) | Gap (%) |
|---|---|---|
| ERM (GD) | 0.00 | 97.71 |
| ERM (GD+M) | 0.00 | 97.71 |
| Warmup+All (Reset) | 67.69 | 31.18 |
| Warmup+All | 74.24 | 24.76 |
| PDE (Reset) | 92.51 | 2.29 |
| PDE | **93.01** | **0.03** |

overall test accuracy comparable to ERM. Furthermore, as depicted in Figure 3(a), ERM rapidly learns the spurious feature as it minimizes the training loss, while barely learning the core feature. Meanwhile, in Figure 3(b) we show the learning of ERM when the data distribution breaks the conditions of our theory and has $\beta_c > \beta_s$ instead. Even with the same $\widehat{\alpha}$ as in Figure 3(a), ERM correctly learns the core feature despite the imbalanced group size. These two figures support the theoretical results we discussed to motivate our method. Consequently, on the same training dataset as in Figure 3(a), Figure 3(c) shows that our approach allows the model to initially learn the core feature using the warm-up dataset and continue learning when incorporating new data.

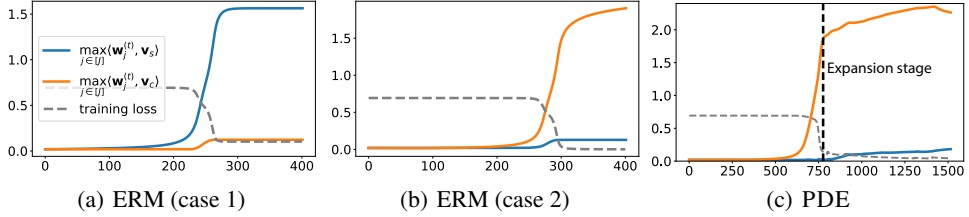

(a) ERM (case 1)        (b) ERM (case 2)        (c) PDE

Figure 3: **Training process of ERM vs. PDE.** We consider the same dataset generated from the distribution as in Definition 2.1 for ERM (case 1) and PDE. On the same training data, ERM learns the spurious feature while PDE successfully learns the core feature. We further consider ERM (case 2) when training on the data distribution where $\beta_c > \beta_s$ and $\widehat{\alpha} = 0.98$. We show the growth of the max inner product between the model's neuron and core/spurious signal vector and the decrease of training loss with regard to the number of iterations $t$.

### 4.2 Real Data

We conduct experiments on real benchmark datasets to: (1) compare our approach with state-of-the-art methods, highlighting its superior performance and efficiency, and (2) offer insights into the design of our method through ablation studies.

**Datasets.** We evaluate on three wildly used datasets across vision and language tasks for spurious correlation: (1) **Waterbirds** (Sagawa et al., 2019) contains bird images labeled as waterbird or landbird, placed against a water or land background, where the smallest subgroup is waterbirds on land background. (2) **CelebA** (Liu et al., 2015) is used to study gender as the spurious feature for hair color classification, and the smallest group in this task is blond-haired males. (3) **CivilComments-WILDS** (Koh et al., 2021b) classifies toxic and non-toxic online comments while dealing with demographic information. It creates 16 overlapping groups for each of the 8 demographic identities.

**Baselines.** We compare our proposed algorithm against several state-of-the-art methods. Apart from standard ERM, we include GroupDRO (Sagawa et al., 2019) and DFR (Kirichenko et al., 2023) that assume access to the group labels. We only include DFR[Tr] for fair comparison as all methods only use the training dataset to train and finetune the model, while DFR[Val] also finetunes on validation data. We also design a baseline called subsample that simply trains the model on the warm-up dataset only. Additionally, we evaluate three recent methods that address spurious correlations without the need for group labels: LfF (Nam et al., 2020), EIIL (Creager et al., 2021), and JTT (Liu et al., 2021). We

Table 2: The worst-group and average accuracy (%) of PDE compared with state-of-the-art methods. The **bold** numbers indicate the best results among the methods that *require group information*, while the underscored numbers represent methods that *only train once*. All methods use validation data for early stopping and model selection, while $\sqrt{}\sqrt{}$ indicates that the method also re-trains the last layer using the validation data

| Method | Group info | Train once | Val info | Waterbirds | | CelebA | | CivilComments | |
|---|---|---|---|---|---|---|---|---|---|
| | | | | Worst | Average | Worst | Average | Worst | Average |
| ERM | $\times$ | $\sqrt{}$ | $\sqrt{}$ | $70.0_{\pm2.3}$ | $97.1_{\pm0.1}$ | $45.0_{\pm1.5}$ | $94.8_{\pm0.2}$ | $58.2_{\pm2.8}$ | $92.2_{\pm0.1}$ |
| LfF | $\times$ | $\times$ | $\sqrt{}$ | $78.0_{N/A}$ | $91.2_{N/A}$ | $77.2_{N/A}$ | $85.1_{N/A}$ | $58.8_{N/A}$ | $92.5_{N/A}$ |
| EIIL | $\times$ | $\times$ | $\sqrt{}$ | $77.2_{\pm1.0}$ | $96.5_{\pm0.2}$ | $81.7_{\pm0.8}$ | $85.7_{\pm0.1}$ | $67.0_{\pm2.4}$ | $90.5_{\pm0.2}$ |
| JTT | $\times$ | $\times$ | $\sqrt{}$ | $86.7_{N/A}$ | $93.3_{N/A}$ | $81.1_{N/A}$ | $88.0_{N/A}$ | $69.3_{N/A}$ | $91.1_{N/A}$ |
| Subsample | $\sqrt{}$ | $\sqrt{}$ | $\sqrt{}$ | $86.9_{\pm2.3}$ | $89.2_{\pm1.2}$ | $86.1_{\pm1.9}$ | $91.3_{\pm0.2}$ | $64.7_{\pm7.8}$ | $83.7_{\pm3.4}$ |
| DFR$^{Tr}$ | $\sqrt{}$ | $\times$ | $\sqrt{}$ | $90.2_{\pm0.8}$ | $97.0_{\pm0.3}$ | $80.7_{\pm2.4}$ | $90.6_{\pm0.7}$ | $58.0_{\pm1.3}$ | $92.0_{\pm0.1}$ |
| DFR$^{Val}$ | $\sqrt{}$ | $\times$ | $\sqrt{}\sqrt{}$ | $\mathbf{92.9}_{\pm0.2}$ | $94.2_{\pm0.4}$ | $88.3_{\pm1.1}$ | $91.3_{\pm0.3}$ | $70.1_{\pm0.8}$ | $87.2_{\pm0.3}$ |
| GroupDRO | $\sqrt{}$ | $\sqrt{}$ | $\sqrt{}$ | $86.7_{\pm0.6}$ | $93.2_{\pm0.5}$ | $86.3_{\pm1.1}$ | $92.9_{\pm0.3}$ | $69.4_{\pm0.9}$ | $89.6_{\pm0.5}$ |
| PDE | $\sqrt{}$ | $\sqrt{}$ | $\sqrt{}$ | $\underline{90.3}_{\pm0.3}$ | $92.4_{\pm0.8}$ | $\mathbf{91.0}_{\pm0.4}$ | $92.0_{\pm0.6}$ | $\mathbf{71.5}_{\pm0.5}$ | $86.3_{\pm1.7}$ |

Table 3: Training efficiency of PDE and GroupDRO on Waterbirds. We compare with GroupDRO at their learning rate and weight decay, as well as at ours. We report the worst-group accuracy, average accuracy and the number of epochs till early stopping as the model reached the best performance on validation data. Note: for a fair comparison, we consider one training epoch as training over the $N$ data as the size of the training dataset.

| Method | Learning rate | Weight decay | Worst | Average | Early-stopping epoch* |
|---|---|---|---|---|---|
| GroupDRO | 1e-5 | 1e-0 | $86.7_{\pm0.6}$ | $93.2_{\pm0.5}$ | $92_{\pm4}$ |
| GroupDRO | 1e-2 | 1e-2 | $77.3_{\pm2.0}$ | $97.1_{\pm0.5}$ | $15_{\pm15}$ |
| PDE | 1e-2 | 1e-2 | $\mathbf{90.3}_{\pm0.3}$ | $92.4_{\pm0.8}$ | $8.9_{\pm1.8}$ |

report results for ERM, Subsample, GroupDRO and PDE based on our own runs using the WILDS library Koh et al. (2021a); for others, we directly reuse their reported numbers.

We present the experiment details including dataset statistics and hyperparameters as well as comprehensive additional experiments in Appendix B.

### 4.2.1 Consistent Superior Worst-group Performance

We assess PDE on the mentioned datasets with the state-of-the-art methods. Importantly, we emphasize the comparison with GroupDRO, as it represents the best-performing method that utilizes group information. As shown in Table 2, PDE considerably enhances the worst-performing group's performance across all datasets, while maintaining the average accuracy comparable to GroupDRO. Remarkably, although GroupDRO occasionally fails to surpass other methods, PDE's performance consistently exceeds them in worst-group accuracy.

### 4.2.2 Efficient Training

In this subsection, we show that our method is more efficient as it does not train a model twice (as in JTT) and more importantly avoids the necessity for a small learning rate (as in GroupDRO). Specifically, methods employing group-balanced batches like GroupDRO require a very small learning rate coupled with a large weight decay in practice. We provide an intuitive explanation as follows. When sampling to achieve balanced groups in each batch, smaller groups appear more frequently than larger ones. If training progresses rapidly, the loss on smaller groups will be minimized quickly, while the majority of the large group data remains unseen and contributes to most of the gradients in later batches. Therefore, these methods necessitate slow training to ensure the model encounters diverse data from larger groups before completely learning the smaller groups. We validate this observation in Table 3, where GroupDRO trained faster than the default results in significantly poorer performance similar to ERM. Conversely, PDE can be trained to converge rapidly on the warm-up set and reaches better worst-group accuracy $10\times$ faster than GroupDRO at default. Note that methods which only finetune the last layer (Kirichenko et al., 2023; Wei et al., 2023) are also efficient. However, they still require training a model first using ERM on the entire training data till convergence. In contrast, PDE does not require to further finetune the model.

### 4.2.3 Understanding the Two Stages

We examine each component and demonstrate their effect in PDE. In Table 4, we present the worst-group and average accuracy of the model trained following the warm-up and expansion stages. Indeed, the majority of learning occurs in the warm-up stage, during which a satisfactory worst-group accuracy is established. In the expansion stage, the model persists in learning new data along the established trajectory, leading to continued performance improvement. In Figure 5,

Table 4: Performance of PDE after each stage. We report the worst-group and average accuracy.

| Dataset | Warm-up | | Addition | |
|---|---|---|---|---|
| | Worst | Avg | Worst | Avg |
| Waterbirds | 86.0 | 91.9 | **90.3** | 92.4 |
| CelebA | 87.8 | 92.1 | **91.0** | 92.0 |
| CivilComm | 67.7 | 78.8 | **71.5** | 86.3 |

we corroborate and emphasize that the model has acquired the appropriate features and maintains learning based on its historical gradient stored in the momentum by presenting the following evidence. As shown, if the optimizer is reset after the warm-up stage and loses all its historical gradients, it rapidly acquires spurious features, resulting in a swift decline in performance accuracy as shown in the blue line.

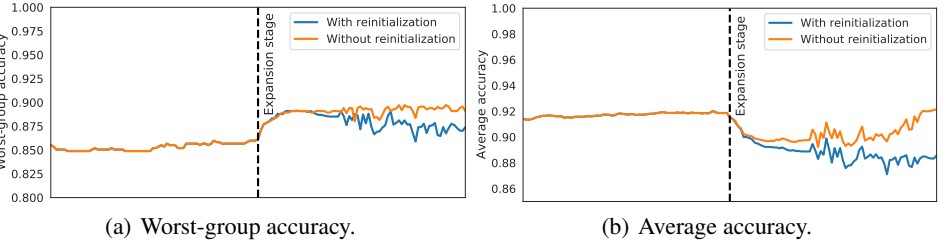

(a) Worst-group accuracy.          (b) Average accuracy.

Figure 4: The effect of reset the momentum to zero after the warm-up stage for PDE on Waterbirds.

### 4.2.4 Ablation Study on the Hyper-parameters of PDE

PDE is robust within a reasonable range of hyperparemeter choices, although some configurations outperform others. As shown in Table 5, it is necessary to limit the number of data points introduced during each expansion to prevent performance degradation. Similarly, in Appendix A, we emphasize the importance of gradual data expansion. In Table 6, we show that post-warmup learning rate decay is essential, though PDE exhibits a tolerance to the degree of this decay. Lastly, as illustrated in Figure 5, adopting a smaller learning rate often necessitates increased data expansions. Nonetheless, a reduced learning rate does not necessarily lead to improved performance.

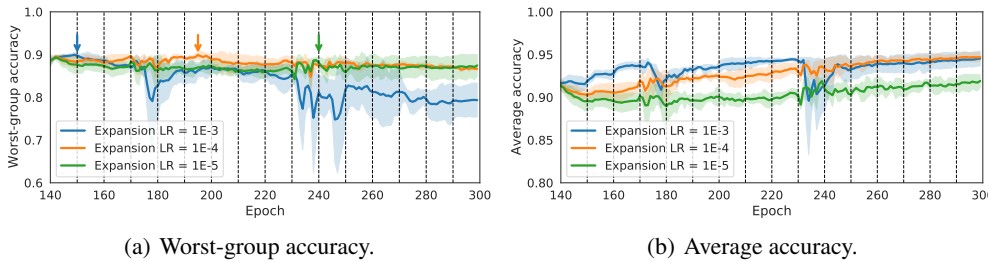

(a) Worst-group accuracy.          (b) Average accuracy.

Figure 5: The variations in both worst-group and average accuracy on the test set of Waterbirds during expansion stage under different expansion learning rates. Each vertical dashed line denotes an expansion and the arrow denotes the early stopping.

Table 5: Ablation study on Waterbirds. Exp. size: number of data points added in each expansion.

| Exp. size | Exp. lr | Worst | Average |
|---|---|---|---|
| 5 | 1e-4 | $89.9_{\pm 0.5}$ | $92.1_{\pm 0.3}$ |
| 10 | 1e-4 | $\mathbf{90.3}_{\pm 0.3}$ | $92.4_{\pm 0.8}$ |
| 50 | 1e-4 | $88.1_{\pm 0.8}$ | $93.4_{\pm 0.4}$ |

Table 6: Ablation study on Waterbirds. Exp. lr: the learning rate in expansion stage.

| Exp. size | Exp. lr | Worst | Average |
|---|---|---|---|
| 10 | 1e-2 | $85.4_{\pm 3.1}$ | $92.1_{\pm 2.0}$ |
| 10 | 1e-3 | $89.4_{\pm 0.7}$ | $92.6_{\pm 0.3}$ |
| 10 | 1e-5 | $89.5_{\pm 0.2}$ | $92.1_{\pm 0.1}$ |

# 5 Related Work

Existing approaches for improving robustness against spurious correlations can be categorized into two lines of research based on the tackled subproblems. A line of research focuses on the same subproblem we tackle: effectively using the group information to improve robustness. With group information, one can use the distributionally robust optimization (DRO) framework and dynamically increase the weight of the worst-group loss in minimization (Hu et al., 2018; Oren et al., 2019; Sagawa et al., 2019; Zhang et al., 2021). Within this line of work, GroupDRO (Sagawa et al., 2019) achieves state-of-the-art performances across multiple benchmarks. Other approaches use importance weighting to reweight the groups (Shimodaira, 2000; Byrd & Lipton, 2019; Xu et al., 2021) and class balancing to downsample the majority or upsample the minority (He & Garcia, 2009; Cui et al., 2019; Sagawa et al., 2020). Alternatively, Goel et al. (2021) leverage group information to augment the minority groups with synthetic examples generated using GAN. Another strategy (Cao et al., 2019, 2020) involves imposing Lipschitz regularization around minority data points. Most recently, methods that train a model using ERM first and then only finetune the last layer on balanced data from training or validation (Kirichenko et al., 2023) or learn post-doc scaling adjustments (Wei et al., 2023) are shown to be effective.

The other line of research focuses on the setting where group information is not available during training and tackles the first subproblem we identified as accurately finding the groups. Recent notable works (Nam et al., 2020; Liu et al., 2021; Creager et al., 2021; Zhang et al., 2022; Yang et al., 2023a) mostly involve training two models, one of which is used to find group information. To finally use the found groups, many approaches (Namkoong & Duchi, 2017; Duchi et al., 2019; Oren et al., 2019; Sohoni et al., 2020) still follow the DRO framework.

The first theoretical analysis of spurious correlation is provided by Sagawa et al. (2020). For self-supervised learning, Chen et al. (2020) shows that fine-tuning with pre-trained models can reduce the harmful effects of spurious features. Ye et al. (2022) provides guarantees in the presence of label noise that core features are learned well only when less noisy than spurious features. These theoretical work only provides analyses of linear models. Meanwhile, a parallel line of work has established theoretical analysis of nonlinear CNNs in the more realistic setting Allen-Zhu & Li (2020); Zou et al. (2021); Wen & Li (2021); Chen et al. (2022); Jelassi & Li (2022). Our work builds on this line of research and generalizes it to the study of spurious features. Lastly, we notice that a concurrent work (Chen et al., 2023) also uses tensor power method (Allen-Zhu & Li, 2020) to analyze the learning of spurious features v.s. invariant features, but in the setting of out-of-distribution generalization.

# 6 Conclusion

In conclusion, this paper addressed the challenge of spurious correlations in training deep learning models and focused on the most effective use of group information to improve robustness. We provided a theoretical analysis based on a simplified data model and a two-layer nonlinear CNN. Building upon this understanding, we proposed PDE, a novel training algorithm that effectively and efficiently enhances model robustness against spurious correlations. This work contributes to both the theoretical understanding and practical application of mitigating spurious correlations, paving the way for more reliable and robust deep learning models.

**Limitations and future work.** Although beyond the linear setting, our analysis still focuses on a relatively simplified binary classification data model. To better represent real-world application scenarios, future work could involve extending to multi-class classification problems and examining the training of transformer architectures. Practically, our proposed method requires the tuning of additional hyperparameters, including the number of warm-up epochs, the number of times for dataset expansion and the number of data to be added in each expansion.

## Acknowledgements

We sincerely thank Dongruo Zhou for the constructive suggestions on the structure and writings of the paper. We also thank the anonymous reviewers for their helpful comments. YD and QG are supported in part by the National Science Foundation CAREER Award 1906169 and IIS-2008981, and the Sloan Research Fellowship. BM is supported by the National Science Foundation CAREER Award 2146492. The views and conclusions contained in this paper are those of the authors and should not be interpreted as representing any funding agencies.

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
