Worst | Waterbirds Average | CelebA Worst | CelebA Average | CivilComments Worst | CivilComments Average |
|---|---|---|---|---|---|---|---|---|---|
| ERM | × | √ | √ | $70.0_{\pm2.3}$ | $97.1_{\pm0.1}$ | $45.0_{\pm1.5}$ | $94.8_{\pm0.2}$ | $58.2_{\pm2.8}$ | $92.2_{\pm0.1}$ |
| LfF | × | × | √ | $78.0_{N/A}$ | $91.2_{N/A}$ | $77.2_{N/A}$ | $85.1_{N/A}$ | $58.8_{N/A}$ | $92.5_{N/A}$ |
| EIIL | × | × | √ | $77.2_{\pm1.0}$ | $96.5_{\pm0.2}$ | $81.7_{\pm0.8}$ | $85.7_{\pm0.1}$ | $67.0_{\pm2.4}$ | $90.5_{\pm0.2}$ |
| JTT | × | × | √ | $86.7_{N/A}$ | $93.3_{N/A}$ | $81.1_{N/A}$ | $88.0_{N/A}$ | $69.3_{N/A}$ | $91.1_{N/A}$ |
| Subsample | √ | √ | √ | $86.9_{\pm2.3}$ | $89.2_{\pm1.2}$ | $86.1_{\pm1.9}$ | $91.3_{\pm0.2}$ | $64.7_{\pm7.8}$ | $83.7_{\pm3.4}$ |
| DFR$^{\text{Tr}}$ | √ | × | √ | $90.2_{\pm0.8}$ | $97.0_{\pm0.3}$ | $80.7_{\pm2.4}$ | $90.6_{\pm0.7}$ | $58.0_{\pm1.3}$ | $92.0_{\pm0.1}$ |

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

## A  Synthetic Experiments

**Datasets.** We generate $10,000$ training examples and $10,000$ test examples from the data distribution defined in Definition 2.1 with dimension $d = 50$ and number of patches $P = 3$. Specifically, we let $\alpha = 0.98$, $\beta_c = 0.2$, $\beta_s = 1$ and $\sigma_p = 0.78$ for Table 1 as well as Figure 3(a) and Figure 3(c). For Figure 3(b), we consider a data distribution where $\alpha = 0.98$, $\beta_c = 1$, $\beta_s = 0.2$ and $\sigma_p = 0.78$. Furthermore, we randomly shuffle the order of the patches of $\mathbf{x}$ after we generate data $(\mathbf{x}, y, a)$.

**Training.** We consider the performances of a nonlinear CNN trained with ERM and PDE. The nonlinear CNN architecture follows (2.3) with the cubic activation function, where we let the number of neurons/filters $J = 40$. We use gradient descent with momentum (GD+M) as the optimizer of our method, setting the momentum to $0.9$ and the learning rate to $0.03$. The number of warm-up iterations is set to $800$. We consider ERM trained with GD with a learning rate $0.1$ and without momentum to align with our theoretical finding in both Table 1 and Figure 3. In Table 1, we also show the experiment results for ERM trained with GD+M as same as PDE. All models are trained until convergence.

**Additional experiments.** In Figure 6, we demonstrate the growth of $\max_{j \in [J]} \langle \mathbf{w}_j^{(t)}, \mathbf{v}_s \rangle$ and $\max_{j \in [J]} \langle \mathbf{w}_j^{(t)}, \mathbf{v}_c \rangle$ for ERM trained with GD+M under the same data generated in Figure 3. Similarly, we observe that ERM learns the spurious feature quickly as the training loss is minimized under our data distribution. Meanwhile, if the data is generated as in case 2 where $\beta_c > \beta_s$, ERM learns the core feature correctly.

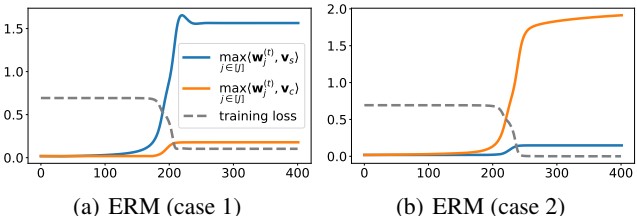

(a) ERM (case 1)          (b) ERM (case 2)

Figure 6: **Training process of ERM trained with GD+M.** We consider the same dataset generated in Figure 3 and observe almost the same training process as ERM with GD, except GD+M learns the features faster.

Furthermore, we consider the following variation of our methods on the same dataset in Table 1 to demonstrate the importance of gradual expansion. In Figure 7, we let PDE to incorporate all of the new training data at once after warm-up stage. As demonstrated, adding all data at once makes it harder for the model to continue learning core feature, resulting in a worst-group accuracy of $74.24\%$ as compared to $94.32\%$ for progressive expansion.

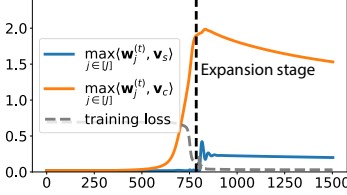

Figure 7: **Variation of PDE.** We consider the same dataset generated in Figure 3 and add all data at once after the warm-up stage.

## B  Benchmark Datasets

**Waterbirds.** The Waterbirds dataset (Sagawa et al., 2019) was constructed to study object recognition models relying on image backgrounds instead of the object itself. To this end, bird images from the Caltech-UCSD Birds-200-2011 (CUB) dataset (Wah et al., 2011) were combined with backgrounds from the Places dataset (Zhou et al., 2017). The dataset contains $4,795$ bird images labeled as a waterbird or landbird and placed against a water or land background. Waterbirds are predominantly located against a water background, while landbirds are situated against a land background. Notably, the smallest subgroup in the dataset is waterbirds on land, consisting of only $56$ examples.

Table 7: Number of data in our wamr-up dataset for PDE's results in Table 2. We also report the number of data in total for the three datasets.

| Dataset | Warm-up | All |
|---|---|---|
| Waterbirds | 224 | 4,795 |
| CelebA | 5,548 | 162,770 |
| CivilComments-WILDS | 13,705 | 269,038 |

**CelebA.** The CelebA dataset (Liu et al., 2015) is a popular face attribute dataset used to examine the spurious associations between non-demographic and demographic attributes. Specifically, one of the 40 binary attributes, "blond hair", is used as the target attribute, and "male" is the spurious attribute. The dataset contains $162,770$ training examples, with the smallest group being blond-haired males, with only 1387 examples.

**CivilComments-WILDS.** The CivilComments-WILDS dataset (Koh et al., 2021b) is designed to explore the challenge of classifying online comments as either toxic or non-toxic while dealing with the spurious correlation between the label and demographic information such as gender, race, religion, and sexual orientation. The dataset's evaluation metric, as defined by Koh et al. (2021b), creates 16 overlapping groups for each of the eight demographic identities, resulting in a total of $512$ distinct groups. For each group, the metric calculates the worst-case performance of a classifier, which allows for a robust evaluation of the model's ability to generalize across diverse populations.

## C   Real Data Experiments

**Setup.** Our experiment settings strictly follow the same setting used for datasets introduced in Appendix B in previous works (Sagawa et al., 2019; Liu et al., 2021; Nam et al., 2020; Creager et al., 2021; Kirichenko et al., 2023). Specifically, we built our training pipeline with the WILDS package (Koh et al., 2021a) which uses pretrained ResNet-50 model (He et al., 2016) in Pytorch (Paszke et al., 2019) library for the image datasets (i.e., Waterbirds and CelebA) and Transformer (Vaswani et al., 2017) in Transformers library (Wolf et al., 2020) for CivilComments-WILDS. All experiments were conducted on a single NVIDIA RTX A6000 GPU with 48GB memory.

**Training.** In Table 7, we summarize the number of data used in warm-up stage for PDE in Table 2 with the total number of data in the entire datasets. In Table 8, we report the hyperparameters used for PDE with the notations in Algorithm 1. Specifically, $T_0$ refers to the number of epochs for warm-up stage and $J$ refers to the number of epochs for training after each data expansion. Lastly, $m$ is the number of added data for each data expansion. Our batch size is consistent with GroupDRO.

Table 8: Hyperparameters used for PDE's results in Table 2. Note that $T_0$ and $J$ are in epochs of PDE's training set, which have fewer iterations than epochs of the full training set.

| Dataset | Learning rate | Weight decay | Batch size | $T_0$ | $J$ | $m$ |
|---|---|---|---|---|---|---|
| Waterbirds | 1e-2 | 1e-2 | 64 | 140 | 10 | 10 |
| CelebA | 1e-2 | 1e-4 | 128 | 16 | 10 | 50 |
| CivilComments-WILDS | 1e-5 | 1e-2 | 16 | 15 | 2 | 300 |

**Groups for CivilComments-WILDS.** We note that the demographic tags in CivilComments-WILDS can coexist in the input text. For example, a text can contain both tags of female and male. Therefore, combining the 8 demographic tags with the binary classification label (toxic vs. non-toxic) results in 16 overlapping groups, where each group counts as data from a class with/without a specific tag. For computational efficiency, previous methods divide the data into four non-overlapping groups either by the *specific* one demographic tag $a_i$ (groups are $\{y = \pm 1, a_i = \pm 1\}$) (Koh et al., 2021b) or by containing *any* one of the tags: $a = 1$ if any $a_i = 1$ and $a = -1$ otherwise (groups are $\{y = \pm 1, a = \pm 1\}$) (Liu et al., 2021; Creager et al., 2021). However, the data can actually be partitioned into $512$ distinct groups, with each group corresponding to different combinations of tags: $\{y = \pm 1, a_1 = \pm 1, a_2 = \pm 1, \ldots, a_n = \pm 1\}$. As GroupDRO requires computation per group at each training batch, considering a large number of groups makes it harder for GroupDRO to train efficiently. Meanwhile, having more groups does not impose an additional computational cost on PDE, so we can consider all these data groups when constructing our warm-up set. As many groups

are empty or contain very little data, we set a threshold to select at most 150 data points from each group to ensure a balanced yet sufficient warm-up set.

**Efficiency.** In Table 9, we further report the training efficiency of PDE compared with GroupDRO on CelebA and CivilComments-WILDS. As similar to what we observe on Waterbirds dataset, PDEachieves best performance at a larger learning rate and smaller weight decay on CelebA with a significant speedup as compared to GroupDRO. On CivilComments-WILDS, we can also observe an improved efficiency.

Table 9: Training efficiency of PDE and GroupDRO on CelebA dataset.

| Method | Learning rate | Weight decay | Worst | Average | Early-stopping epoch* |
|---|---|---|---|---|---|
| GroupDRO | 1e-5 | 1e-1 | $86.3_{\pm 1.1}$ | $92.9_{\pm 0.3}$ | $23.7_{\pm 6.8}$ |
| PDE | 1e-2 | 1e-4 | $\mathbf{91.0}_{\pm 0.4}$ | $92.0_{\pm 0.6}$ | $\mathbf{0.7}_{\pm 0.3}$ |

Table 10: Training efficiency of PDE and GroupDRO on CivilComments-WILDS dataset.

| Method | Learning rate | Weight decay | Worst | Average | Early-stopping epoch* |
|---|---|---|---|---|---|
| GroupDRO | 1e-5 | 1e-2 | $69.4_{\pm 0.9}$ | $89.6_{\pm 0.5}$ | $3.3_{\pm 2.1}$ |
| PDE | 1e-5 | 1e-2 | $\mathbf{71.5}_{\pm 0.5}$ | $86.3_{\pm 1.7}$ | $\mathbf{2.1}_{\pm 1.1}$ |

**Data Augmentation.** Additionally, the increased training speed of our method facilitates the usage of techniques such as data augmentation. While data augmentation is a common practice for improving model generalization, DRO approaches have not incorporated it into their methods. We hypothesize that this omission stems from the slower training process. Data augmentation introduces random noise to the training data, which complicates convergence during training when using a very small learning rate. As illustrated in Table 11, data augmentation leads to slightly worse performance for GroupDRO. In contrast, our method effectively benefits from data augmentation.

Table 11: The effect of data augmentation on GroupDRO and PDE on Waterbirds dataset. We report the worst-group and average accuracy.

| | GroupDRO | | PDE | |
|---|---|---|---|---|
| Method | Worst | Avg | Worst | Avg |
| W/o data aug | 86.7 | 93.2 | 88.9 | 89.5 |
| W/ data aug | 85.7 | 96.6 | **90.3** | **92.4** |

# D  Proof Preliminaries

**Notation.** In this paper, we use lowercase letters, lowercase boldface letters, and uppercase boldface letters to respectively denote scalars ($a$), vectors ($\mathbf{v}$), and matrices ($\mathbf{W}$). We use sgn to denote the sign function.For a vector $\mathbf{v}$, we use $\|\mathbf{v}\|_2$ to denote its Euclidean norm. Given two sequences $\{x_n\}$ and $\{y_n\}$, we denote $x_n = \mathcal{O}(y_n)$ if $|x_n| \leq C_1 |y_n|$ for some absolute positive constant $C_1$, $x_n = \Omega(y_n)$ if $|x_n| \geq C_2 |y_n|$ for some absolute positive constant $C_2$, and $x_n = \Theta(y_n)$ if $C_3 |y_n| \leq |x_n| \leq C_4 |y_n|$ for some absolute constants $C_3, C_4 > 0$. We use $\widetilde{\mathcal{O}}(\cdot)$ to hide logarithmic factors of $d$ in $\mathcal{O}(\cdot)$.

Before we go into the analysis, we first consider the following gradient,

$$\nabla_{\mathbf{w}_j}\mathcal{L}(\mathbf{W}^{(t)}) = -\frac{1}{N}\sum_{i=1}^{N}\frac{\exp(-y_i f(\mathbf{x}_i; \mathbf{W}^{(t)}))}{1 + \exp(-y_i f(\mathbf{x}_i; \mathbf{W}^{(t)}))} \cdot y_i f'(\mathbf{x}_i; \mathbf{W}^{(t)}). \tag{D.1}$$

Let's denote the derivative of a data example $i$ at iteration $t$ to be

$$\ell_i^{(t)} = \frac{\exp(-y_i f(\mathbf{x}_i; \mathbf{W}^{(t)}))}{1 + \exp(-y_i f(\mathbf{x}_i; \mathbf{W}^{(t)}))} = \text{sigmoid}(-y_i f(\mathbf{x}_i; \mathbf{W}^{(t)})). \tag{D.2}$$

**Lemma D.1.** (Gradient) Let the loss function $\mathcal{L}$ be as defined in (2.1). For $t \geq 0$ and $j \in [J]$, the gradient of the loss $\mathcal{L}$ with regard to neuron $\mathbf{w}_j$ is

$$\nabla_{\mathbf{w}_j}\mathcal{L}(\mathbf{W}^{(t)}) = -\frac{3}{N}\bigg(\beta_c^3 \sum_{i=1}^{N} \ell_i^{(t)} \langle \mathbf{w}_j, \mathbf{v}_c \rangle^2 \mathbf{v}_c + \sum_{i=1}^{N} \ell_i^{(t)} y_i \langle \mathbf{w}_j, \boldsymbol{\xi}_i \rangle^2 \boldsymbol{\xi}_i +$$

$$\left(\sum_{i \in S_1} \ell_i^{(t)} - \sum_{i \in S_2} \ell_i^{(t)}\right) \cdot \beta_s^3 \langle \mathbf{w}_j, \mathbf{v}_s \rangle^2 \mathbf{v}_s\right).$$

*Proof.* We have the following gradient

$$\nabla_{\mathbf{w}_j} \mathcal{L}(\mathbf{W}^{(t)}) = -\frac{1}{N} \sum_{i=1}^{N} \frac{\exp(-y_i f(\mathbf{x}_i; \mathbf{W}^{(t)}))}{1 + \exp(-y_i f(\mathbf{x}_i; \mathbf{W}^{(t)}))} \cdot y_i f'(\mathbf{x}_i; \mathbf{W}^{(t)}). \tag{D.3}$$

And let's denote the derivative of a data example $i$ at iteration $t$ to be

$$\ell_i^{(t)} = \frac{\exp(-y_i f(\mathbf{x}_i; \mathbf{W}^{(t)}))}{1 + \exp(-y_i f(\mathbf{x}_i; \mathbf{W}^{(t)}))} = \text{sigmoid}(-y_i f(\mathbf{x}_i; \mathbf{W}^{(t)})). \tag{D.4}$$

Then, we can further write the gradient as

$$\begin{aligned}
\nabla_{\mathbf{w}_j} \mathcal{L}(\mathbf{W}^{(t)}) &= -\frac{3}{N} \sum_{i=1}^{N} \ell_i^{(t)} y_i \sum_{p=1}^{P} \langle \mathbf{w}_j, \mathbf{x}^{(p)} \rangle^2 \cdot \mathbf{x}^{(p)} \\
&= -\frac{3}{N} \sum_{i=1}^{N} \ell_i^{(t)} y_i \Big( \langle \mathbf{w}_j, \beta_c y_i \mathbf{v}_c \rangle^2 \beta_c y_i \mathbf{v}_c + \langle \mathbf{w}_j, \beta_s a_i \mathbf{v}_s \rangle^2 \beta_s a_i \mathbf{v}_s + \langle \mathbf{w}_j, \boldsymbol{\xi}_i \rangle^2 \boldsymbol{\xi}_i \Big) \\
&= -\frac{3}{N} \sum_{i=1}^{N} \ell_i^{(t)} \Big( \beta_c^3 \langle \mathbf{w}_j, \mathbf{v}_c \rangle^2 \mathbf{v}_c + \beta_s^3 y_i a_i \langle \mathbf{w}_j, \mathbf{v}_s \rangle^2 \mathbf{v}_s + y_i \langle \mathbf{w}_j, \boldsymbol{\xi}_i \rangle^2 \boldsymbol{\xi}_i \Big) \\
&= -\frac{3}{N} \Bigg( \sum_{i=1}^{N} \ell_i^{(t)} \Big( \beta_c^3 \langle \mathbf{w}_j, \mathbf{v}_c \rangle^2 \mathbf{v}_c + y_i \langle \mathbf{w}_j, \boldsymbol{\xi}_i \rangle^2 \boldsymbol{\xi}_i \Big) \\
&\qquad + \Big( \sum_{i \in S_1} \ell_i^{(t)} - \sum_{i \in S_2} \ell_i^{(t)} \Big) \beta_s^3 \langle \mathbf{w}_j, \mathbf{v}_s \rangle^2 \mathbf{v}_s \Bigg),
\end{aligned}$$

where the last equality holds due to that for $i \in S_1$ we have $a_i = y_i$ and for $i \in S_2$ we have $a_i = -y_i$. $\qquad\square$

With the gradient, we have the following:

**Core feature gradient.** The projection of the gradient on $\mathbf{v}_c$ is then

$$\langle \nabla_{\mathbf{w}_j} \mathcal{L}(\mathbf{W}^{(t)}), \mathbf{v}_c \rangle = -\frac{3\beta_c^3}{N} \sum_{i=1}^{N} \ell_i^{(t)} \langle \mathbf{w}_j, \mathbf{v}_c \rangle^2. \tag{D.5}$$

**Spurious feature gradient.** The projection of the gradient on $\mathbf{v}_s$ is

$$\langle \nabla_{\mathbf{w}_j} \mathcal{L}(\mathbf{W}^{(t)}), \mathbf{v}_s \rangle = -\frac{3\beta_s^3}{N} \Big( \sum_{i \in S_1} \ell_i^{(t)} - \sum_{i \in S_2} \ell_i^{(t)} \Big) \cdot \langle \mathbf{w}_j, \mathbf{v}_s \rangle^2. \tag{D.6}$$

**Noise gradient.** The projection of the gradient on $\boldsymbol{\xi}_i$ is

$$\langle \nabla_{\mathbf{w}_j} \mathcal{L}(\mathbf{W}^{(t)}), \boldsymbol{\xi}_i \rangle = -\frac{3y_i}{N} \sum_{i=1}^{N} \ell_i^{(t)} \langle \mathbf{w}_j, \boldsymbol{\xi}_i \rangle^2 \|\boldsymbol{\xi}_i\|_2^2. \tag{D.7}$$

**Derivative of data example $i$.** $\ell_i^{(t)}$ can be rewritten as

$$\begin{aligned}
\ell_i^{(t)} &= \text{sigmoid}\big( -y_i f(\mathbf{x}_i; \mathbf{W}^{(t)}) \big) \\
&= \text{sigmoid}\Big( \sum_{j=1}^{J} -\beta_c^3 \langle \mathbf{w}_j, \mathbf{v}_c \rangle^3 - y_i a_i \beta_s^3 \langle \mathbf{w}_j, \mathbf{v}_s \rangle^3 - y_i \langle \mathbf{w}_j, \boldsymbol{\xi}_i \rangle^3 \Big). \tag{D.8}
\end{aligned}$$

Note that $0 < \ell_i^{(t)} < 1$ due to the property of the sigmoid function. Furthermore, we similarly consider that the sum of the sigmoid terms for all time steps is bounded up to a logarithmic dependence (Chen et al., 2022). And the sigmoid term is considered small for a $\kappa$ such that

$$\sum_{t=0}^{T} \frac{1}{1 + \exp(\kappa)} \le \widetilde{O}(1),$$

which implies $\kappa \ge \widetilde{\Omega}(1)$.

## E  Proof of Theorem 2.2

In this section, we present the detailed proofs that build up to Theorem 2.2. We begin by considering the update for spurious feature and core feature.

**Lemma E.1** (Spurious feature update.). For all $t \ge 0$ and $j \in [J]$, the spurious feature update is

$$\langle \mathbf{w}_j^{(t+1)}, \mathbf{v}_s \rangle = \langle \mathbf{w}_j^{(t)}, \mathbf{v}_s \rangle + \frac{3\eta\beta_s^3}{N} \Big( \sum_{i \in S_1} \ell_i^{(t)} - \sum_{i \in S_2} \ell_i^{(t)} \Big) \langle \mathbf{w}_j^{(t)}, \mathbf{v}_s \rangle^2,$$

which gives

$$\widetilde{\Theta}(\eta)\beta_s^3 \Big( \widehat{\alpha}g_1(t) - \sum_{i \in S_2} \ell_i^{(t)}/N \Big) \langle \mathbf{w}_j^{(t)}, \mathbf{v}_s \rangle^2 \le \langle \mathbf{w}_j^{(t+1)}, \mathbf{v}_s \rangle - \langle \mathbf{w}_j^{(t)}, \mathbf{v}_s \rangle$$

$$\le \widetilde{\Theta}(\eta)\beta_s^3 \cdot \widehat{\alpha}g_1(t) \cdot \langle \mathbf{w}_j^{(t)}, \mathbf{v}_s \rangle^2,$$

where $g_1(t) = \mathrm{sigmoid}\big( - \sum_{j \in [J]} (\beta_c^3 \langle \mathbf{w}_j^{(t)}, \mathbf{v}_c \rangle^3 + \beta_s^3 \langle \mathbf{w}_j^{(t)}, \mathbf{v}_s \rangle^3) \big)$.

*Proof.* The spurious feature update is obtained by using the gradient update of $\mathbf{W}^{(t)}$ and plugging in (D.6):

$$\langle \mathbf{w}_j^{(t+1)}, \mathbf{v}_s \rangle = \langle \mathbf{w}_j^{(t)} - \eta\nabla_{\mathbf{w}_j}\mathcal{L}(\mathbf{W}^{(t)}), \mathbf{v}_s \rangle$$

$$= \langle \mathbf{w}_j^{(t)}, \mathbf{v}_s \rangle + \frac{3\eta\beta_s^3}{N} \Big( \sum_{i \in S_1} \ell_i^{(t)} - \sum_{i \in S_2} \ell_i^{(t)} \Big) \langle \mathbf{w}_j^{(t)}, \mathbf{v}_s \rangle^2.$$

We first prove for the upper bound. Consider the following,

$$\langle \mathbf{w}_j^{(t+1)}, \mathbf{v}_s \rangle = \langle \mathbf{w}_j^{(t)}, \mathbf{v}_s \rangle + \frac{3\eta\beta_s^3}{N} \Big( \sum_{i \in S_1} \ell_i^{(t)} - \sum_{i \in S_2} \ell_i^{(t)} \Big) \langle \mathbf{w}_j^{(t)}, \mathbf{v}_s \rangle^2$$

$$\le \langle \mathbf{w}_j^{(t)}, \mathbf{v}_s \rangle + \frac{3\eta\beta_s^3}{N} \Big( \sum_{i \in S_1} \ell_i^{(t)} \Big) \langle \mathbf{w}_j^{(t)}, \mathbf{v}_s \rangle^2$$

$$\le \langle \mathbf{w}_j^{(t)}, \mathbf{v}_s \rangle + \widetilde{\Theta}(\eta)\beta_s^3 \cdot \frac{\sum_{i \in S_1} g_1(t)}{N} \cdot \langle \mathbf{w}_j^{(t)}, \mathbf{v}_s \rangle^2$$

$$= \langle \mathbf{w}_j^{(t)}, \mathbf{v}_s \rangle + \widetilde{\Theta}(\eta)\beta_s^3 \widehat{\alpha} \cdot g_1(t) \cdot \langle \mathbf{w}_j^{(t)}, \mathbf{v}_s \rangle^2,$$

where the first inequality holds due to $0 < \ell_i^{(t)} < 1$, the second inequality holds due to Lemma G.4, and the last equality holds due to $|S_1|/N = \widehat{\alpha}$. Then, for the lower bound, we consider the same bound for $i \in S_1$ in Lemma G.4 and obtain

$$\langle \mathbf{w}_j^{(t+1)}, \mathbf{v}_s \rangle = \langle \mathbf{w}_j^{(t)}, \mathbf{v}_s \rangle + \frac{3\eta\beta_s^3}{N} \Big( \sum_{i \in S_1} \ell_i^{(t)} - \sum_{i \in S_2} \ell_i^{(t)} \Big) \langle \mathbf{w}_j^{(t)}, \mathbf{v}_s \rangle^2$$

$$\ge \langle \mathbf{w}_j^{(t)}, \mathbf{v}_s \rangle + \widetilde{\Theta}(\eta)\beta_s^3 \Big( \widehat{\alpha} \cdot g_1(t) - \sum_{i \in S_2} \ell_i^{(t)}/N \Big) \langle \mathbf{w}_j^{(t)}, \mathbf{v}_s \rangle^2.$$

$\square$

Similarly, we have the update for core feature as below.

**Lemma E.2** (Core feature update). For all $t \geq 0$ and $j \in [J]$, the core feature update is

$$\langle \mathbf{w}_j^{(t+1)}, \mathbf{v}_c \rangle = \langle \mathbf{w}_j^{(t)}, \mathbf{v}_c \rangle + \frac{3\eta\beta_c^3}{N} \Big( \sum_{i=1}^{N} \ell_i^{(t)} \Big) \langle \mathbf{w}_j^{(t)}, \mathbf{v}_c \rangle^2,$$

which gives

$$\widetilde{\Theta}(\eta)\beta_c^3 \cdot \widehat{\alpha} g_1(t) \cdot \langle \mathbf{w}_j^{(t)}, \mathbf{v}_c \rangle^2 \leq \langle \mathbf{w}_j^{(t+1)}, \mathbf{v}_c \rangle - \langle \mathbf{w}_j^{(t)}, \mathbf{v}_c \rangle$$
$$\leq \widetilde{\Theta}(\eta)\beta_c^3 \cdot \Big( \widehat{\alpha} g_1(t) + \sum_{i \in S_2} \ell_i^{(t)}/N \Big) \cdot \langle \mathbf{w}_j^{(t)}, \mathbf{v}_c \rangle^2.$$

*Proof.* The core feature update is obtained by using the gradient update of $\mathbf{W}^{(t)}$ and plugging in (D.5):

$$\langle \mathbf{w}_j^{(t+1)}, \mathbf{v}_c \rangle = \langle \mathbf{w}_j^{(t)} - \eta \nabla_{\mathbf{w}_j} \mathcal{L}(\mathbf{W}^{(t)}), \mathbf{v}_c \rangle$$
$$= \langle \mathbf{w}_j^{(t)}, \mathbf{v}_c \rangle + \frac{3\eta\beta_c^3}{N} \Big( \sum_{i=1}^{N} \ell_i^{(t)} \Big) \langle \mathbf{w}_j^{(t)}, \mathbf{v}_c \rangle^2.$$

We prove for the lower bound,

$$\langle \mathbf{w}_j^{(t+1)}, \mathbf{v}_c \rangle = \langle \mathbf{w}_j^{(t)}, \mathbf{v}_c \rangle + \frac{3\eta\beta_c^3}{N} \Big( \sum_{i=1}^{N} \ell_i^{(t)} \Big) \langle \mathbf{w}_j^{(t)}, \mathbf{v}_c \rangle^2$$
$$\geq \langle \mathbf{w}_j^{(t)}, \mathbf{v}_c \rangle + \frac{3\eta\beta_c^3}{N} \Big( \sum_{i \in S_1} \ell_i^{(t)} \Big) \langle \mathbf{w}_j^{(t)}, \mathbf{v}_c \rangle^2$$
$$\geq \langle \mathbf{w}_j^{(t)}, \mathbf{v}_c \rangle + \widetilde{\Theta}(\eta)\beta_c^3 \widehat{\alpha} g_1(t) \cdot \langle \mathbf{w}_j^{(t)}, \mathbf{v}_c \rangle^2,$$

where the first inequality holds due to $0 < \ell_i^{(t)} < 1$ and the second inequality holds due to Lemma G.4. And for the upper bound. we similarly have

$$\langle \mathbf{w}_j^{(t+1)}, \mathbf{v}_c \rangle = \langle \mathbf{w}_j^{(t)}, \mathbf{v}_c \rangle + \frac{3\eta\beta_c^3}{N} \Big( \sum_{i=1}^{N} \ell_i^{(t)} \Big) \langle \mathbf{w}_j^{(t)}, \mathbf{v}_c \rangle^2$$
$$\leq \langle \mathbf{w}_j^{(t)}, \mathbf{v}_c \rangle + \widetilde{\Theta}(\eta)\beta_c^3 \cdot \Big( \widehat{\alpha} g_1(t) + \sum_{i \in S_2} \ell_i^{(t)} \Big) \cdot \langle \mathbf{w}_j^{(t)}, \mathbf{v}_c \rangle^2.$$

$\square$

Note that $\langle \mathbf{w}_j^{(t+1)}, \mathbf{v}_c \rangle$ is non-decreasing from the lower bound of Lemma E.2. As $\mathbf{w}_j^{(0)} \sim \mathcal{N}(0, \sigma_0^2 \mathbf{I}_d)$ are initialized with small $\sigma_0$, the sigmoid terms $\ell_i^{(t)}$ are large in the initial iterations. And while $l_i^{(t)}$ remains large for $i \in S_1$, we have $g_1(t) = \Theta(1)$ as similar as in Jelassi & Li (2022). Therefore, $\langle \mathbf{w}_j^{(t+1)}, \mathbf{v}_s \rangle$ is also non-decreasing since $\widehat{\alpha} \cdot \Theta(1) - \sum_{i \in S_2} l_i^{(t)}/N \geq 2\widehat{\alpha} - 1 > 0$ for $l_i^{(t)} < 1$ and $\widehat{\alpha} > 1/2$. Eventually, $g_1(t)$ becomes small at a time $T_0 > 0$. We now consider a simplified version of the above lemma in this early training stage.

**Lemma E.3** (Spurious feature update in early iterations). Let $T_0 > 0$ be such that $\max_{j \in [J]} \langle \mathbf{w}_j^{(T_0)}, \mathbf{v}_s \rangle \geq \widetilde{\Omega}(1/\beta_s)$. For $t \in [0, T_0]$, the spurious feature update has the following bound

$$\widetilde{\Theta}(\eta)\beta_s^3(2\widehat{\alpha} - 1) \cdot \langle \mathbf{w}_j^{(t)}, \mathbf{v}_s \rangle^2 \leq \langle \mathbf{w}_j^{(t+1)}, \mathbf{v}_s \rangle - \langle \mathbf{w}_j^{(t)}, \mathbf{v}_s \rangle \leq \widetilde{\Theta}(\eta)\beta_s^3 \widehat{\alpha} \cdot \langle \mathbf{w}_j^{(t)}, \mathbf{v}_s \rangle^2.$$

*Proof.* Let $T_0 > 0$ be such that either $\max_{j \in [J]} \langle \mathbf{w}_j^{(T_0)}, \mathbf{v}_s \rangle \geq \widetilde{\Omega}(1/\beta_s)$ or $\max_{j \in [J]} \langle \mathbf{w}_j^{(T_0)}, \mathbf{v}_c \rangle \geq \widetilde{\Omega}(1/\beta_c)$. We will show later that the first condition will be first met and we have $\langle \mathbf{w}_j^{(t)}, \mathbf{v}_c \rangle \leq \widetilde{\Omega}(1/\beta_c)$ for all $j \in [J]$ and $t \in [0, T_0]$.

Recall that $g_1(t) = \text{sigmoid}\big(-\sum_{j \in [J]}(\beta_c^3 \langle \mathbf{w}_j^{(t)}, \mathbf{v}_c \rangle^3 + \beta_s^3 \langle \mathbf{w}_j^{(t)}, \mathbf{v}_s \rangle^3)\big)$. Then, for $t \in [0, T_0]$, we have

$$g_1(t) = \frac{1}{1 + \exp\big(\sum_{j \in [J]}(\beta_c^3 \langle \mathbf{w}_j^{(t)}, \mathbf{v}_c \rangle^3 + \beta_s^3 \langle \mathbf{w}_j^{(t)}, \mathbf{v}_s \rangle^3)\big)}$$

$$\geq \frac{1}{1 + \exp(\kappa + \kappa)}$$

$$= \frac{1}{1 + \exp(\widetilde{\Omega}(1))},$$

where the first inequality holds due to $\langle \mathbf{w}_s^{(t)}, \mathbf{v}_s \rangle \leq \kappa/(J^{1/3}\beta_s)$ and $\langle \mathbf{w}_s^{(t)}, \mathbf{v}_c \rangle \leq \kappa/(J^{1/3}\beta_c)$ for $t \in [0, T_0]$. Therefore, similar to Jelassi & Li (2022), we have $g_1(t) = \Theta(1)$ in the early iterations. Moreover, as $0 < \ell_i^{(t)} < 1$, we have $\sum_{i \in S_2} \ell_i^{(t)}/N < 1 - \widehat{\alpha}$. This implies the result in Lemma E.1 as

$$\widetilde{\Theta}(\eta)\beta_s^3(2\widehat{\alpha} - 1)\langle \mathbf{w}_j^{(t)}, \mathbf{v}_s \rangle^2 \leq \langle \mathbf{w}_j^{(t+1)}, \mathbf{v}_s \rangle - \langle \mathbf{w}_j^{(t)}, \mathbf{v}_s \rangle \leq \widetilde{\Theta}(\eta)\beta_s^3\widehat{\alpha}\langle \mathbf{w}_j^{(t)}, \mathbf{v}_s \rangle^2.$$

$\square$

And similarly for core feature, we have

**Lemma E.4** (Core feature update in early iterations). Let $T_0 > 0$ be such that $\max_{j \in [J]} \langle \mathbf{w}_j^{(T_0)}, \mathbf{v}_s \rangle \geq \widetilde{\Omega}(1/\beta_s)$. For $t \in [0, T_0]$, the core feature update has the following bound

$$\widetilde{\Theta}(\eta)\beta_c^3\widehat{\alpha} \cdot \langle \mathbf{w}_j^{(t)}, \mathbf{v}_c \rangle^2 \leq \langle \mathbf{w}_j^{(t+1)}, \mathbf{v}_c \rangle - \langle \mathbf{w}_j^{(t)}, \mathbf{v}_c \rangle \leq \widetilde{\Theta}(\eta)\beta_c^3 \cdot \langle \mathbf{w}_j^{(t)}, \mathbf{v}_c \rangle^2.$$

*Proof.* Let $T_0 > 0$ be such that either $\max_{j \in [J]} \langle \mathbf{w}_j^{(T_0)}, \mathbf{v}_s \rangle \geq \widetilde{\Omega}(1/\beta_s)$ or $\max_{j \in [J]} \langle \mathbf{w}_j^{(T_0)}, \mathbf{v}_c \rangle \geq \widetilde{\Omega}(1/\beta_c)$. Again, with $g_1(t) = \Theta(1)$ and $\sum_{i \in S_2} \ell_i^{(t)}/N < 1 - \widehat{\alpha}$ as shown in Lemma E.3, we can imply the result in Lemma E.2 as

$$\widetilde{\Theta}(\eta)\beta_c^3\widehat{\alpha} \cdot \langle \mathbf{w}_j^{(t)}, \mathbf{v}_c \rangle^2 \leq \langle \mathbf{w}_j^{(t+1)}, \mathbf{v}_c \rangle - \langle \mathbf{w}_j^{(t)}, \mathbf{v}_c \rangle \leq \widetilde{\Theta}(\eta)\beta_c^3 \cdot \langle \mathbf{w}_j^{(t)}, \mathbf{v}_c \rangle^2,$$

which completes the proof. $\square$

With the updates of spurious and core feature in the early iterations, we can now show with the following lemma that GD will learn the spurious feature very quickly while hardly learns the core feature.

**Lemma E.5.** Let $T_0$ be the iteration number that $\max_{j \in [J]} \langle \mathbf{w}_j^{(t)}, \mathbf{v}_s \rangle$ reaches $\widetilde{\Omega}(1/\beta_s) = \widetilde{\Theta}(1)$. Then, we have for all $t \leq T_0$, it holds that $\max_{j \in [J]} \langle \mathbf{w}_j^{(t)}, \mathbf{v}_c \rangle = \widetilde{O}(\sigma_0)$.

*Proof.* Consider the following from Lemma E.3 and Lemma E.4,

$$\langle \mathbf{w}_j^{(t+1)}, \mathbf{v}_c \rangle - \langle \mathbf{w}_j^{(t)}, \mathbf{v}_c \rangle \leq \widetilde{\Theta}(\eta)\beta_c^3 \cdot \langle \mathbf{w}_j^{(t)}, \mathbf{v}_c \rangle^2$$

$$\langle \mathbf{w}_j^{(t+1)}, \mathbf{v}_s \rangle - \langle \mathbf{w}_j^{(t)}, \mathbf{v}_s \rangle \geq \widetilde{\Theta}(\eta)\beta_s^3(2\widehat{\alpha} - 1)\langle \mathbf{w}_j^{(t)}, \mathbf{v}_s \rangle^2.$$

Recall that we initialize the weights as $\mathbf{w}_j^{(0)} \sim \mathcal{N}(\mathbf{0}, \sigma_0^2)$. We have $\langle \mathbf{w}_j^{(0)}, \mathbf{v}_c \rangle \sim \mathcal{N}(0, \sigma_0^2)$ and $\langle \mathbf{w}_j^{(0)}, \mathbf{v}_s \rangle \sim \mathcal{N}(0, \sigma_0^2)$. For the weights have small initialization with $\sigma_0 = \text{polylog}(d)/d$, we have $O(\langle \mathbf{w}_j^{(0)}, \mathbf{v}_c \rangle) = O(\langle \mathbf{w}_j^{(0)}, \mathbf{v}_s \rangle)$. Therefore, for $\beta_c^3 = o(1)$ and $\beta_s^3(2\widehat{\alpha} - 1) = \Theta(1)$, we call Lemma G.1 and get

$$\langle \mathbf{w}_j^{(T_0)}, \mathbf{v}_c \rangle \leq O(\langle \mathbf{w}_j^{(0)}, \mathbf{v}_s \rangle) = \widetilde{O}(\sigma_o)$$

for all $j \in [J]$. $\square$

Given the above lemma, we can conclude that the condition $\max_{j \in [J]} \langle \mathbf{w}_j^{(T_0)}, \mathbf{v}_s \rangle \geq \widetilde{\Omega}(1/\beta_s)$ will be first met. And therefore, $T_0$ is such that $\max_{j \in [J]} \langle \mathbf{w}_j^{(T_0)}, \mathbf{v}_s \rangle \geq \widetilde{\Omega}(1/\beta_s)$.

**Theorem E.6** (Restatement of Theorem 2.2). Consider the training dataset $S = \{(\mathbf{x}_i, y_i)\}_{i=1}^{N}$ that follows the distribution in Definition 2.1. Consider the two-layer nonlinear CNN model as in (2.3) initialized with $\mathbf{W}^{(0)} \sim \mathcal{N}(0, \sigma_0^2)$. After training with GD in (2.2) for $T_0 = \widetilde{\Theta}\big(1/(\eta\beta_s^3\sigma_0)\big)$ iterations, for all $j \in [J]$ and $t \in [0, T_0)$, we have

$$\widetilde{\Theta}(\eta)\beta_s^3(2\widehat{\alpha} - 1) \cdot \langle \mathbf{w}_j^{(t)}, \mathbf{v}_s \rangle^2 \leq \langle \mathbf{w}_j^{(t+1)}, \mathbf{v}_s \rangle - \langle \mathbf{w}_j^{(t)}, \mathbf{v}_s \rangle \leq \widetilde{\Theta}(\eta)\beta_s^3\widehat{\alpha} \cdot \langle \mathbf{w}_j^{(t)}, \mathbf{v}_s \rangle^2, \quad \text{(E.1)}$$

$$\widetilde{\Theta}(\eta)\beta_c^3\widehat{\alpha} \cdot \langle \mathbf{w}_j^{(t)}, \mathbf{v}_c \rangle^2 \leq \langle \mathbf{w}_j^{(t+1)}, \mathbf{v}_c \rangle - \langle \mathbf{w}_j^{(t)}, \mathbf{v}_c \rangle \leq \widetilde{\Theta}(\eta)\beta_c^3 \cdot \langle \mathbf{w}_j^{(t)}, \mathbf{v}_c \rangle^2. \quad \text{(E.2)}$$

After training for $T_0$ iterations, with high probability, the learned weight has the following properties: (1) it learns the spurious feature $\mathbf{v}_s$: $\max_{j \in [J]} \langle \mathbf{w}_j^{(T)}, \mathbf{v}_s \rangle \geq \widetilde{\Omega}(1/\beta_s)$; (2) it does not learn the core feature $\mathbf{v}_c$: $\max_{j \in [J]} \langle \mathbf{w}_j^{(T)}, \mathbf{v}_c \rangle = \widetilde{\mathcal{O}}(\sigma_0)$.

*Proof.* The updates directly follow the results from Lemma E.1 and Lemma E.2. And the result for $\max_{j \in [J]} \langle \mathbf{w}_j^{(t)}, \mathbf{v}_c \rangle$ follows Lemma E.5. It remains to calculate the time $T_0$. With Lemma G.2, we consider the sequence for $\max_{j \in [J]} \langle \mathbf{w}_j^{(t+1)}, \mathbf{v}_s \rangle$, where by Lemma E.3,

$$\langle \mathbf{w}_j^{(t+1)}, \mathbf{v}_s \rangle \leq \langle \mathbf{w}_j^{(t)}, \mathbf{v}_s \rangle + \widetilde{\Theta}(\eta)\beta_s^3\widehat{\alpha} \cdot \langle \mathbf{w}_j^{(t)}, \mathbf{v}_s \rangle^2,$$

$$\langle \mathbf{w}_j^{(t+1)}, \mathbf{v}_s \rangle \geq \langle \mathbf{w}_j^{(t)}, \mathbf{v}_s \rangle + \widetilde{\Theta}(\eta)\beta_s^3(2\widehat{\alpha} - 1) \cdot \langle \mathbf{w}_j^{(t)}, \mathbf{v}_s \rangle^2.$$

As $\langle \mathbf{w}_j^{(t)}, \mathbf{v}_s \rangle$ is non-decreasing in early iterations and with high probability, there exist an index $j$ such that $\langle \mathbf{w}_j^{(0)}, \mathbf{v}_s \rangle \geq 0$. Among all the possible indices $i \in [J]$ that are initialized to have posive inner product with $\mathbf{v}_s$, we focus on the max index $r = \arg\max_{j \in [J]} \langle \mathbf{w}_j^{(0)}, \mathbf{v}_s \rangle$. Then with $v = \widetilde{\Theta}(1/\beta_s)$ in Lemma G.2, we will have $T_0$ as

$$T_0 = \frac{\widetilde{\Theta}(1)}{\eta\alpha^3\sigma_0} + \frac{\widetilde{\Theta}(1)\widehat{\alpha}}{2\widehat{\alpha} - 1} \left\lceil \frac{-\log\left(\sigma_0\beta_s\right)}{\log(2)} \right\rceil.$$

$\square$

## F   Proof of Lemma 3.1

**Lemma F.1** (Restatement of Lemma 3.1). Given the balanced training dataset $S^0 = \{(\mathbf{x}_i, y_i, a_i)\}_{i=1}^{N_0}$ with $\widehat{\alpha} = 1/2$ as in Definition 2.1 and CNN as in (2.3). The gradient on $\mathbf{v}_s$ will be 0 from the beginning of training.

*Proof.* With Lemma D.1, the projection of the gradient on $\mathbf{v}_s$ in the initial iteration ($t < T_0$) is

$$\langle \nabla_{\mathbf{w}_j} \mathcal{L}(\mathbf{W}^{(t)}), \mathbf{v}_s \rangle = -\frac{3\beta_s^3}{N} \Big( \sum_{i \in S_1} \ell_i^{(t)} - \sum_{i \in S_2} \ell_i^{(t)} \Big) \cdot \langle \mathbf{w}_j, \mathbf{v}_s \rangle^2$$

$$= \Theta\left(\frac{\beta_s^3}{N}\right) \Big( |S_1| - |S_2| \Big)$$

$$= 0,$$

where the first equality is due to $\ell_i^{(t)} = \Theta(1)$ in the initial iterations and the second equality is due to $\widehat{\alpha} = 0.5$. $\square$

## G   Auxiliary Lemmas

**Lemma G.1** (Lemma C.20, Allen-Zhu & Li 2020). Let $\{x_t, y_t\}_{t=1,..}$ be two positive sequences that satisfy

$$x_{t+1} \geq x_t + \eta \cdot Ax_t^2,$$

$$y_{t+1} \leq y_t + \eta \cdot By_t^2,$$

for some $A = \Theta(1)$ and $B = o(1)$. Suppose $y_0 = O(x_0)$ and $\eta < O(x_0)$, and for all $C \in [X_0, O(1)]$, let $T_x$ be the first iteration such that $x_t \geq C$. Then, we have $T_x\eta = \Theta(x_0^{-1})$ and

$$y_{T_x} \leq O(x_0).$$

**Lemma G.2** (Lemma K.15, Jelassi & Li 2022). Let $\{z_t\}_{t=0}^T$ be a positive sequence defined by the following recursions

$$z_{t+1} \geq z_t + m(z_t)^2,$$
$$z_{t+1} \leq z_t + M(z_t)^2,$$

where $z_0 > 0$ is the initialization and $m, M > 0$ are some constants. Let $v > 0$ such that $z_0 \leq v$. Then, the time $t_0$ such that $z_t \geq v$ for all $t \geq t_0$ is

$$t_0 = \frac{3}{mz_0} + \frac{8M}{m}\left\lceil \frac{\log(v/z_0)}{\log(2)} \right\rceil.$$

We make the following assumptions for every $t \leq T$ as the same in (Jelassi & Li, 2022).

**Lemma G.3** (Induction hypothesis D.1, Jelassi & Li 2022). Throughout the training process using GD for $t \leq T$, we maintain that, for every $i \in S_1$ and $j \in [J]$,

$$|\langle \mathbf{w}_j^{(t)}, \boldsymbol{\xi}_i \rangle| \leq \widetilde{O}(\sigma_0 \sigma \sqrt{d}). \tag{G.1}$$

**Lemma G.4.** For $i \in S_1$, we have $\ell_i^{(t)} = \Theta(1)g_1(t)$, where

$$g_1(t) = \text{sigmoid}\Big( - \sum_{j \in [J]}(\beta_c^3 \langle \mathbf{w}_j^{(t)}, \mathbf{v}_c \rangle^3 + \beta_s^3 \langle \mathbf{w}_j^{(t)}, \mathbf{v}_s \rangle^3)\Big).$$

*Proof.* Given $i \in S_1$, we have from (D.8) that

$$\ell_i^{(t)} = \text{sigmoid}\Big( \sum_{j=1}^J -\beta_c^3 \langle \mathbf{w}_j, \mathbf{v}_c \rangle^3 - \beta_s^3 \langle \mathbf{w}_j, \mathbf{v}_s \rangle^3 - y_i \langle \mathbf{w}_j, \boldsymbol{\xi}_i \rangle^3 \Big)$$

$$= 1 \Big/ \Big( 1 + \exp\Big( \sum_{j=1}^J \beta_c^3 \langle \mathbf{w}_j, \mathbf{v}_c \rangle^3 + \beta_s^3 \langle \mathbf{w}_j, \mathbf{v}_s \rangle^3 + y_i \langle \mathbf{w}_j, \boldsymbol{\xi}_i \rangle^3 \Big)\Big). \tag{G.2}$$

Recall induction hypothesis G.3, we have the following for $i \in S_1$,

$$|y_i \langle \mathbf{w}_j^{(t)}, \boldsymbol{\xi}_i \rangle| \leq \widetilde{O}(\sigma_0 \sigma \sqrt{d})$$
$$\iff -\widetilde{O}(\sigma_0 \sigma \sqrt{d}) \leq y_i \langle \mathbf{w}_j^{(t)}, \boldsymbol{\xi}_i \rangle \leq \widetilde{O}(\sigma_0 \sigma \sqrt{d}), \tag{G.3}$$

where $|y_i| = 1$. Plug (G.3) back into (G.2), we get

$$e^{-\widetilde{O}(\sigma_0 \sigma \sqrt{d})^3}g_1(t) \leq \ell_i^{(t)} \leq e^{\widetilde{O}(\sigma_0 \sigma \sqrt{d})^3}g_1(t).$$

With our parameter setting, we have $\widetilde{O}(\sigma_0 \sigma \sqrt{d}) = \widetilde{O}(\sigma_0) = \widetilde{O}(\text{polylog}(d)/d)$. Therefore, $e^{\pm\widetilde{O}(\sigma_0 \sigma \sqrt{d})^3} = \Theta(1)$. □