# OpenReview forum: "Robust Learning with Progressive Data Expansion Against Spurious Correlation"
_NeurIPS.cc/2023/Conference — NeurIPS 2023 poster_

### Official Review · Reviewer_uURe · 2023-06-29

**Soundness:** 3 good
**Presentation:** 3 good
**Contribution:** 3 good
**Rating:** 6
**Confidence:** 4

**Summary:**

The paper describes Progressive Data Expansion (PDE), a new method for training models which are robust to spurious features. The idea of the method is to split the training into two phases: warmup and expansion. In the warmup stage, the dataset is balanced, and the model learns a classifier that ignores the spurious feature. In the expansion stage, the remaining data is used to finetune. The authors also provide a theoretical argument about feature learning, and show strong empirical performance on benchmark datasets.

**W** &mdash; weakness,  **S** &mdash; strength, **Q** &mdash; question.

**Strengths:**

**S1.** The authors provide a theoretical analysis, that provides some intuition for the method
**S2.** The proposed method is simple and computationally cheap
**S3.** The authors achieve strong empirical performance across three benchmark datasets

**Weaknesses:**

**Q1/W1. Relationship to DFR**

The works [1, 2] advocated for an approach that is almost opposite to PDE: they train a model with standard ERM, and then retrain only the last layer on a group-balanced dataset. PDE on the other hand pretrains the model on a group-balanced dataset, and then finetunes with standard ERM. The methods achieve similar performance, if we consider DFR^Val, which is the main method in [1] (the authors argue that DFR^Val uses the validation data, but I believe PDE also uses the same amount of validation data to tune the hyper-parameters?). In fact, PDE achieves better performance on CelebA and CivilComments, while DFR achieves better performance on waterbirds.

However, conceptually [1, 2] argue that in fact standard ERM learns the core features as well as GroupDRO on the standard benchmarks, and once the last layer is retrained we can recover the performance of GroupDRO. This intuition is quite the opposite of the intuition presented in this paper, which is that ERM does not learn the core features without intervention. I think it would be good if the authors could comment on this distinction.

My intuition is that the pretraining in the warmup stage prevents the model from learning the spurious feature, and makes it harder for the model to use the spurious features in the expansion stage. Alternatively, it could be that the model still represents the spurious features, but because of the momentum, it is unable to switch to using this feature in the expansion phase. It would be interesting to explore more deeply which of these options is happening.

Lastly, in line 287 you mention that the method is more efficient than DFR, because it does not train the model twice. However, DFR only retrains the last layer of a pretrained ERM model, which is computationally cheap, so I don't believe this statement is correct.

 **Q2/W2. Robustness to the choice of hyper-parameters**

It is important to understand how sensitive the method is to the choice of hyper-parameters. In particular, I would be interested in the impact of
- Length of the expansion training phase
- Learning rate in the expansion phase

My intuition is that the method continues going in a good direction for some time during the expansion phase, but then eventually it should converge to an optimal solution on the full data distribution, i.e. maximize the average accuracy and not the worst group accuracy. However, due to early stopping, the authors are able to achieve good performance, before the optimization learns to use the spurious feature. Is this intuition correct?

**W3. Theoretical analysis**

The theoretical analysis is quite toy. The authors use a model which is basically a bag of features, i.e. the result is a sum of cubed dot-products of features and "filters". This model is not very non-linear. However, it is understandable, given that theoretical analysis of non-linear models is challenging.


**Questions:**

**Q3. Expansion phase:** In line 225 you mention that during the expansion phase you try to sample data uniformly across groups. What do you do exactly in this phase? Isn't all of the minority data used in the warmup phase?

**Q4. Intuition for expansion phase:** Related to Q3, why is performance on the training data distribution go down in Fig 4 (b), even when you reset the momentum variables? Is the training loss going up then? Why would this happen? Or is the data not just sampled from the training distribution early in the expansion phase?

**Limitations:**

**References**

[1] [_Last Layer Re-Training is Sufficient for Robustness to Spurious Correlations_](https://openreview.net/forum?id=Zb6c8A-Fghk);
P. Kirichenko, P. Izmailov, A. G. Wilson;
ICLR 2023

[2] [_On Feature Learning in the Presence of Spurious Correlations_](https://proceedings.neurips.cc/paper_files/paper/2022/hash/fb64a552feda3d981dbe43527a80a07e-Abstract-Conference.html);
P. Izmailov, P. Kirichenko, N. Gruver, A. G. Wilson;
NeurIPS 2022

---

> ### Author Rebuttal · Authors · 2023-08-10
>
> We appreciate your positive feedback and suggestions that helped us improving our work. We hope our clarifications answer your questions.
>
> ---
>
> **Q1a**: Consider DFR^Val.
>
> **A1a**: Thanks for pointing this out. We agree that all methods use the information from validation data for hyper-parameter tuning and model selection, including PDE and DFR^Tr. Yet, there is still a difference for DFR^Val that it also **trains the last layer on validation data**, which is updating the model parameters in addition to hyper-parameters. For a comprehensive comparison, we've updated our table (as in Table 2 of the attached PDF) to include DFR^Val, with a notation to distinguish whether the validation data is used solely for hyperparameter tuning or also for updating the final layer of the model itself.
>
> **Q1b**: The intuition for DFR is quite the opposite of the intuition in this paper.
>
> **A1b**: We believe that PDE and DFR approach the problem differently but don't contradict each other in their understandings. Our theory indicates that, for ERM, the learning of spurious features will quickly overshadow that of core features, resulting in the reliance on spurious features when making predictions. As in Theorem 2.2, the learning of spurious feature will have surpassed a threshold while the learning of core feature remains at the same scale as initialization.
>
> That said, we don't intend to assert that ERM won't learn the core feature at all. Despite the dominance of spurious feature in prediction, the model will still learn the core feature at a minimal rate compared to spurious. We believe that there is no contradiction between us and the intuition of DFR to recover and amplify this learning of core feature through later interventions. To reflect this, we will revise the wording of Theorem 2.2 to make it more rigorous.
>
> However, in PDE, the case is very different as spurious feature in balanced dataset is not useful and its gradient remains 0. The model only tends to learn spurious feature in the expansion stage if without any further intervention, as the expansion data is not group-balanced. Hence, we need momentum from warmup, so that we use the core feature preserved in momentum to amplify the core feature gradient in expansion data so that it continues to dominate the spurious feature. Our arguments are also supported by synthetic experiments. In Figure 3a of our paper, there is an observable minimal learning of core feature for ERM. Meanwhile, in Figure 3c, the model does not learn spurious feature during warmup where the groups are balanced.
>
> **Q1c**: Efficiency comparison with DFR.
>
> **A1c**: We acknowledge and appreciate the efficiency of the fine-tuning approaches. We’ll clarify this in our manuscript to reflect that DFR retrains the last layer only. However, while it's true that re-training the final layer doesn't demand significant additional computation, it still requires first training a model using ERM. Meanwhile, PDE itself converges much faster to its optimal performance as compared to the ERM stage of DFR. **We still firmly believe that our statement on the efficiency of PDE is correct.**
>
> ---
>
> **Q2**: Robustness to the choice of hyper-parameters.
>
> **A2**: We appreciate the suggestion and added ablation studies in Table 4 the PDF. Namely, PDE is robust within a reasonable range of hyperparemeter selections. Meanwhile, there are preferred choices for the hyperparameters.
> - The number of data added during each expansion cannot be too large as it will harm the performance. As similarly demonstrated in Figure 6 of Appendix A, gradual expansion is important to our method and adding all data at once will result in a worse performance.
> - There is a necessity to decay the learning rate after the warmup stage, while PDE is relatively robust to the extent of decay.
> - The number of times for data expansion depends on the early-stopping on the validation data. As in Figure 1 in the PDF, a smaller learning rate will result in more data expansions. However, a smaller learning rate does not necessarily result in a better performance.
>
> We will add the ablation studies and corresponding discussion to the revised version of our work.
>
> ---
>
> **Q3**: Isn't all of the minority data used in the warmup phase?
>
> **A3**: We'd like to clarify our methodology related to the usage of training data, particularly during the warmup stage. In this initial stage, we employ all data from the smallest group in the training dataset (waterbirds, land background), while for all other groups, we select a random subset equivalent in size to this smallest group. Post-warmup, we add additional data from the training dataset that were not seen during the warmup stage. At this point, new data from other small groups still remain (landbirds, water background). If possible, we aim to maintain balance among the remaining groups when enlarging the training dataset. Ultimately, the newly added data will exclusively come from the largest group (landbirds, land background). We will make sure to explain this more clearly in our revision.
>
> ---
>
> **Q4**: Is the data not just sampled from the training distribution early in the expansion phase?
>
> **A4**: To clarify, our method uses the training data (and not the validation set) for warmup and expansion stages, with the intent to include all training data by the end of the expansion stage. While the data used in the expansion phase is drawn from the training dataset, remains new to the model, having not been used during the warmup stage. As we add new data, there's a minor increase in training loss, which quickly drops as training advances. Regarding the referenced figure, both average accuracy and worst-group accuracy are plotted on test data. As we use this figure to highlight the importance of momentum, we want to focus on model's actual performance as opposed to the training. We believe that the performance on test data will not always align with the trend of training loss as well.

---

> > ### Comment · Reviewer_uURe · 2023-08-21
> > **Thank you for the rebuttal!**
> >
> > Dear authors, thank you for the rebuttal! I appreciate the new results on the robustness of performance with respect to the expansion period training length, and other clarifications.

---

### Official Review · Reviewer_YdZz · 2023-07-05

**Soundness:** 3 good
**Presentation:** 2 fair
**Contribution:** 2 fair
**Rating:** 5
**Confidence:** 4

**Summary:**

In this paper, the author propose a learning framework for improving the performance of classifiers on the worst group in the presence of spurious features. The authors focus on a two-layer convolutional neural network. They first show that imbalanced data and easy-to-learn spurious features can lead to bias in the classifier. Then, they propose a learning procedure that focuses on balanced data and gradually progresses how the model learns from core features to all features. The new method is evaluated on several datasets demonstrating that the new approach is fast and can improve accuracy on worst groups.

**Strengths:**

The paper is mostly well-written and easy to follow, and the idea makes sense. The proposed approach is motivated theoretically using a two-layer CNN model. The method is simple and relies on a two-stage training procedure, therefore can be easily implemented and adapted to other algorithms.
The proposed method improves the performance of the worst group in all evaluated examples; in some cases, it leads to significant improvements.


**Weaknesses:**

The theoretical analysis in the paper is focused on binary classification. The method requires tuning several hyperparameters. While the worst group performance is improved in most cases the overall accuracy degradation is non negligible. The method is only evaluated on a small number of datasets.

**Questions:**

How is the method applied to non-binary data?
The analysis focuses on the activation z^3. Can this be replaced with more commonly used activations?
Why is there a significant drop in average accuracy in some examples?
Some parts are not well explained; for example, the caption in Figure 1 is unclear.


**Limitations:**

Limitations are discussed in the last section of the paper. Addressing some of the limitations presented by the authors is indeed important to increase the usability of the proposed approach.

---

> ### Author Rebuttal · Authors · 2023-08-10
>
> We appreciate the feedback and questions raised on the potential confusions. Please find our detailed response below, for which we hope that the reviewer considers increasing their evaluation of our work.
>
> ---
>
> **Q1**: The theoretical analysis in the paper is focused on binary classification. The method requires tuning several hyperparameters.
>
> **A1**: We have indeed recognized the aforementioned points in our limitation section and believe they offer valuable directions for future research. We would like to further emphasize that the previous papers [1-4] also focused on the binary classification problem. To the best of our knowledge, no existing study has studied multi-class classification problems in the theoretical analysis of spurious correlations. Meanwhile, we consider the more complex case of a non-linear CNN, addressing data distributions that conventional linear models find challenging.
>
> We also want to kindly remind the NeurIPS guidelines on the limitation discussion that “authors should be rewarded rather than punished for being up front about the limitations of their work.”
>
> [1] Sagawa et al. "An investigation of why overparameterization exacerbates spurious correlations."
>
> [2] Chen et al. "Self-training avoids using spurious features under domain shift."
>
> [3] Yang et al. "Understanding rare spurious correlations in neural networks."
>
> [4] Ye et al. "Freeze then train: Towards provable representation learning under spurious correlations and feature noise."
>
> ---
>
> **Q2**: While the worst group performance is improved in most cases the overall accuracy degradation is non negligible. Why is there a significant drop in average accuracy in some examples?
>
> **A2**: Thanks for raising the question. We wish to clarify that the research on spurious correlations primarily aims to develop a **robust and reliable predictor**, and therefore **worst-group accuracy is regarded as the main evaluation metric** rather than striving for an even higher average accuracy than ERM. This can be found in previous studies [1-5], which highlighted the importance of improving worst-group accuracy and similarly exhibited such “trade-off”. The state-of-the-art methods like GroupDRO and DFR all improves worst-group accuracy while dropping the average accuracy.
>
> Furthermore, the decrease in the average accuracy is not necessarily considered as a drawback in the context of spurious correlations. The presence of spurious correlations can mislead us into perceiving a high average accuracy as an indicator of a reliable predictor. However, a model may perform exceptionally well on larger groups, yet fail smaller ones. In contrast, the gap between average accuracy and worst-group accuracy, as outlined in [6], can help identify if a model is overly influenced by spurious features. For a more straightforward comparison, we updated a table comparing the gaps for all models in the 1-page pdf in Table 3. The objective of creating a more dependable model is therefore to minimize this gap while enhancing worst-group accuracy.
>
> ---
>
> **Q3**: The method is only evaluated on a small number of datasets.
>
> **A3**: We have included the most common benchmark datasets used for spurious correlations as in previous works [1-6]. Admittedly, the benchmark datasets for spurious correlation are limited, where Waterbirds and CelebA have been the most used data. We believe the development of more benchmark datasets specifically designed to evaluate models' performance on spurious correlations is an important future work. However, within the scope of our current study, we have endeavored to provide a comprehensive evaluation using the most prevalent datasets available.
>
> [1] Sagawa et al. "Distributionally Robust Neural Networks."
>
> [2] Liu et al. "Just train twice: Improving group robustness without training group information."
>
> [3] Creager et al. "Environment inference for invariant learning."
>
> [4] Zhang et al. "Correct-N-Contrast: a Contrastive Approach for Improving Robustness to Spurious Correlations."
>
> [5] Kirichenko et al. "Last Layer Re-Training is Sufficient for Robustness to Spurious Correlations."
>
> [6] Haghtalab at al. "On-demand sampling: Learning optimally from multiple distributions."
>
> ---
>
> **Q4**: The analysis focuses on the activation z^3. Can this be replaced with more commonly used activations?
>
> **A4**: Yes, it indeed can be replaced with the activations such as ReLU, as demonstrated by other works applying a similar analysis but for different problems [1]. The cubic activation has been widely used in the other theoretical works [2] for the simplicity of analysis and results that align with experiments. The inclusion of ReLU activation, while feasible, will make our analysis more intricate, especially in terms of proof details. Our primary goal in this study is to maintain simplicity and ensure our theoretical motivation is easy to follow, thereby providing clear motivation for our proposed method. Lastly, our experiments also confirmed that our results hold in practice for ReLU activations.
>
> [1] Kou, et al. "Benign overfitting for two-layer relu networks."
>
> [2] Jelassi et al. "Towards understanding how momentum improves generalization in deep learning."
>
> ---
>
> We hope that you can revisit your assessment of our work in light of the clarifications provided above. Should there be any other concerns, we are happy to provide further information.

---

> > ### Comment · Reviewer_YdZz · 2023-08-15
> > **Response to authors**
> >
> > I thank the authors for the hard work spent on this rebuttal. I have some comments about the response:
> > Q1) These were stated as weaknesses, and they indeed are; despite appearing in the limitations, these are some of the disadvantages of the method and therefore should be mentioned in this bullet. I truly appreciate that the authors mentioned these and gave them positive credit.
> >
> > Q2) I understand this tradeoff and the importance of improving worst-case performance. Still, DFR offers a better tradeoff based on your reported results. Nonetheless, I see the value of improving performance on the worst case but suggest that this issue be discussed in the paper.
> >
> > Q3) Following the papers mentioned by the authors I see additional datasets including MultiNLI, CMNIST, and Adult-Confounded.
> >
> > Q4) Noted; better to mention this.
> >
> > To conclude, the authors have addressed most of my concerns and I decided to raise my score.

---

> > > ### Author Response · Authors · 2023-08-16
> > > **Thank you!**
> > >
> > > Thank you for your positive feedback and increasing the score! We'll be sure to incorporate all the suggested changes into our final version.  Regarding the additional datasets, we will add experiments on the CMNIST and MultiNLI datasets in our upcoming revision. We'll update the experimental results here if time allows.

---

### Official Review · Reviewer_6r1Q · 2023-07-06

**Soundness:** 4 excellent
**Presentation:** 3 good
**Contribution:** 4 excellent
**Rating:** 8
**Confidence:** 3

**Summary:**

The paper works on understanding and mitigating the impacts on spurious features. Specifically, the authors provide theoretical analysis on the learning process of a non-linear two-layer CNN, under spurious features. Theoretical insights reveal the need to start with balanced data, and progressively expand the train set. The proposed PDE is right based on such insights, and is demonstrated to be efficient and effective on certain datasets.

**Strengths:**

* (1) The authors provide theoretical insights on the learning process of a two-layer non-linear CNN under spurious features, which aligns with empirical observations.

* (2) The proposed method starts with group-balanced training set and then gradually injects more data samples to capture the core features. And its effectiveness and efficiency are demonstrated on several popular datasets when compared with exiting methods.

* (3) The paper is clearly presented and well organized, and insights/motivations are easy to follow.



**Weaknesses:**

Although the proposed PDE is super efficient by referring to existing learning/training based approaches, it would be better to mention/discuss the efficiency of some fine-tuning or post-hoc approaches during the related work section, for example, fine-tune the last layer is sufficient to spurious features [R1], post-hoc adjustment on model prediction improves robustness to spurious features [R2].

References:

R1: Last Layer Re-Training is Sufficient for Robustness to Spurious Correlations. [ICLR'23]

R2: Distributionally Robust Post-hoc Classifiers under Prior Shifts. [ICLR'23]

**Questions:**

* (1) The results on the synthetic data is a bit confusing: in table 1, the performance of ERM on worst-group is 0%; besides, PDE has a better worst-group performance than the overall one, which is kind of conterintuitive.

* (2) It would be much better if authors could add more discussions on the role and selection/suggestions of certain parameters, i.e., the number of times for dataset expansion and the number of data to be added in each expansion.

**Limitations:**

The authors clearly stated their limitations at the end of paper,

* The theoretical analysis rely on simplified model;

* Certain hyper-parameters might be crucial in model training: i.e., the number of warmup epochs, the number of times for dataset expansion and the number of data to be added in each expansion.

---

> ### Author Rebuttal · Authors · 2023-08-10
>
> We're grateful for your strong support and suggestions on our work, for which we have accordingly made modifications to our manuscript. Please find our detailed response below for the questions raised in the review.
>
> ---
>
> **Q1**: It would be better to mention/discuss the efficiency of some fine-tuning or post-hoc approaches during the related work section.
>
> **A1**: We thank the reviewer for pointing out the post-hoc approaches and we will add a discussion in the related work section in our revision. Briefly, we believe that fine-tuning approaches [R1, R2] that re-trains the last layer are also efficient methods that can effectively mitigate the influence of spurious correlations. Nevertheless, these methods still require training a model first using ERM in the first stage and finetune the last layer in the second stage. For example, on the Waterbirds dataset, the model is first trained for 300 epochs on the entire training dataset using ERM and further finetuned in the second stage. As we train PDE with no more than 9 epochs as compared to ERM, we believe PDE can be more efficient since PDE also does not require to further finetune the model. Nevertheless, we believe that for [R1, R2], finetuning the last layer only does not incur significant additional computation to ERM.
>
> ---
>
> **Q2**: The results on the synthetic data is a bit confusing.
>
> **A2**: Thanks for pointing out the inconsistency of PDE’s result in Table 1. We apologize for the oversight in reporting the incorrect worst-group results; the value we presented was actually for the small group accuracy. This has now been corrected, and the table has been updated accordingly in Table 1 of the attached PDF.
>
> The 0% accuracy of ERM is accurate. Our synthetic data was specifically designed so that a model only relying on spurious features will completely fail the small-group test data. If the model leans heavily on the spurious feature, its predictions will align with the spurious features that are the opposite to the true labels in the small groups, resulting in worse accuracy than random guessing. Admittedly, the synthetic setting is a simpler and more extreme version of the real data. In real-world datasets, ERM does achieve 45.0% worst-group accuracy on CelebA and 58.2% on CivilComments.
>
> ---
>
> **Q3**. Add more discussions on the role and selection/suggestions of certain parameters.
>
> **A3**. We appreciate the suggestion and added ablation studies on Waterbirds dataset as in Table 4 and Figure 1 of the attached PDF. Namely, our methods are robust within a reasonable range of hyperparameter selections. Meanwhile, there are preferred choices for the hyperparameters.
> - The number of data added during each expansion cannot be too large as it will harm the performance. As similarly demonstrated in Figure 6 of Appendix A, gradual expansion is important to our method and adding all data at once will result in a worse performance.
> - There is a necessity to decay the learning rate after the warmup stage, while PDE is relatively robust to the extent of decay.
> - The number of times for data expansion depends on the early-stopping on the validation data. As in Figure 1 in the PDF, a smaller learning rate will result in more data expansions.
>
> We will add the ablation studies and corresponding discussion to the revised version of our work.

---

> > ### Comment · Reviewer_6r1Q · 2023-08-21
> >
> > Thanks authors for the detailed responses and ablation studies, which have addressed most of my concerns. After reading the author's rebuttal and the other reviewers' comments, I would like to keep my original score (8: Strong Accept).

---

### Official Review · Reviewer_vLux · 2023-07-07

**Soundness:** 2 fair
**Presentation:** 3 good
**Contribution:** 2 fair
**Rating:** 4
**Confidence:** 5

**Summary:**

The authors propose a new method called PDE to address spurious features and improve generalization. Building on the existing literature, the authors consider an imbalanced binary classification problem where two types of features coexist: core features and spurious features. The core features can lead to good generalization performance but a classification model may pick up spurious features. To avoid that, the proposed method (1) starts with a downsampled balanced classification as a warm-up, which will not pick up the spurious features, and (2) then gradually increase the imbalance ratio until the full dataset is used for training.

The main justification for this two-stage algorithm is that gradient descent with momentum can benefit from the warm-up stage where no spurious features are picked up. Numerous experiments on synthetic data and real data are used to support the proposed method.

**Strengths:**

Classification with spurious features is a very practical and relevant problem. The proposed method is easy to implement and new to my knowledge. The experiments result show promising performance compared with alternatives.

Overall, the warm-up stage in optimization seems to be an interesting idea. It starts with an "easy" downsampled dataset without data imbalance---which is known to be the culprit of picking up spurious features. Then, in the second stage, we hope to use the full datasets without picking up spurious features by using momentum.

The presentation of this paper is clear and theoretical/experimental results are easy to follow.

**Weaknesses:**

In my opinion, this paper misses a crucial element in classification with spurious features---that is, the effect of overparametrization. It is known, for example in [1], that there is a very important difference between underparameterization and overparametrization when spurious features are present. It is, therefore, expected to have analyses on the inductive bias of the proposed method and how it may select a superior solution compared with ERM and recent alternatives. Unfortunately, the effect of overparametrization is absent in the analyses, as the number of neurons $J$ does not play a role in the analysis.

Besides, while the two-stage optimization idea looks interesting, the analysis is a bit handwaving, so it is unconvincing to me why momentum method is able to keep avoiding spurious features in the second stage. An investigation of inductive bias would make this paper much stronger; perhaps even some in-depth experiments on synthetic datasets would shed lights on the behavior of momentum in terms on avoiding spurious features.

Lastly, I think the experimental results look fine, but I am not convinced that the proposed method is better than the recent alternatives. For example, in Table 2, while PDE achieves better accuracy on worst classes, it usually incurs a worse accuracy on average. Since there is a clear tradeoff between majority and minority classes in binary classification, better accuracy for the minority class can be often trivially improved by shifting the decision boundary. It would be more convincing if empirical results show improvements for *all* classes.


[1] Sagawa, S., Raghunathan, A., Koh, P. W., and Liang, P. An investigation of why overparameterization exacerbates spurious correlations. In International Conference on Machine Learning, pp. 8346– 8356. PMLR, 2020.

**Questions:**

See the weaknesses section.

---

> ### Author Rebuttal · Authors · 2023-08-10
>
> We thank the reviewer for the feedback. We believe that there are some misunderstandings of the contribution of our paper. In light of our clarifications below, we hope that the reviewer considers increasing their evaluation of our work.
>
> ---
>
> **Q1**: This paper misses the effect of overparameterization.
>
> **A1**: We believe there is a misunderstanding of our results. We'd like to clarify that the CNN considered in our analysis is indeed overparameterized. Specifically, "overparameterization" refers to settings where a model's parameters exceed the data's dimension. In our study, the parameters in our CNNs amount to $J \times d$ which is indeed larger than the data dimension. As similar to previous work [2], the order of $J$ is assumed to be $J=\text{polylog}(d)$ and the model has a mild overparameterization. We will emphasize this point more explicitly in the revision.
>
> The fundamental work [1] studies the effect of spurious correlations on underparametrized model compared to overparameterized model, and confirmed that overparametrized models are specifically negatively affected by spurious correlations. Hence, we do not investigate the parameterization effect again but focus specifically on the overparameterized regime. **Based on their initial findings, we aim to further understand what properties of data cause the learning of spurious features and consequently how to avoid it for overparameterized models.**  The major focus of PDE is to improve the robustness of current deep learning models against spurious correlations. Hence, we do not investigate the underparameterized models that are less used in practice and proven not affected. We want to highlight that, not only in our analysis, but also all models considered in our synthetic and real experiments are overparameterized. Lastly, as many related theory papers explore different aspects of analysis under the overparameterized setting, we especially focus on the non-linear setting and consider convolutional architectures for the first time.
>
> ---
>
> **Q2**:  It is unconvincing to me why momentum method is able to keep avoiding spurious features in the second stage.
>
> **A2**: To show the importance of momentum, we have dedicated line 210-228 for explanation and real data experiments in Figure 4. Briefly, the role of momentum that preserves historical trend of learning is proven in the previous work [2]. We use this theoretical finding to motivate our method. As discussed, the model learns the core feature in the warmup stage. During expansion, the momentum from warmup amplifies the core feature that is also present in the gradients of newly added data. This learning process will then correspond to the second case when model learns core feature discussed in subsection 3.1, and the model can tolerate the imbalanced expansion data and continues learning of core feature.
>
> Furthermore in Figure 4, our findings suggest that re-initializing the optimizer, and consequently its historical gradient after the warmup stage, will harm the model performance compared to preserving momentum from the warmup stage. Moreover, we provide more ablation studies on synthetic data in Table 1 in the 1-page PDF. As shown, keeping the momentum from the warmup stage provides effective improvement to the worst-group accuracy of the model.
>
> ---
>
> **Q3**: Not convinced that PDE is better than the recent alternatives. It would be more convincing if empirical results show improvements for all classes.
>
> **A3**: We clarify that **worst-group accuracy is the main objective for studies in spurious correlation, and has been regarded as the main evaluation criteria in prior work**. As evident in prior works [3-6], the focus has consistently been on enhancing worst-group accuracy. A high average accuracy can often be misleading, as it might suggest a dependable predictor while the model relies on spurious correlations. Consequently, the predictor performs remarkably well on larger groups but fail when the spurious correlation is not present or indicate the contrary.
>
> We also believe there is a misunderstanding of the evaluation metric. Worst-group accuracy is not computed with regard to the **classes** and a simple "shift of the decision boundary" is not practical. Instead, the **groups** represent varying combinations of spurious signals (backgrounds) and true labels (birds). Small groups are present within all classes, such as waterbirds with land background and landbirds with water background. **They are not a single under-performing class.** Hence, class accuracy is similar to average accuracy and doesn't indicate reliability in this context.
>
> We are happy to include an additional evaluation metric here. While a high average accuracy does not indicate a reliable predictor but can in fact be alarming, we include here another evaluation metric as per prior literature [7]. This metric measures the gap between the worst-group and average accuracy, where a smaller gap indicates a more robust model that's potentially less reliant on the spurious correlation. The results are presented in Table 3 in the PDF.
>
> ---
>
> [1] Sagawa et al. "An investigation of why overparameterization exacerbates spurious correlations."
>
> [2] Jelassi et al. "Towards understanding how momentum improves generalization in deep learning."
>
> [3] Liu et al. "Just train twice: Improving group robustness without training group information."
>
> [4] Creager et all. "Environment inference for invariant learning."
>
> [5] Zhang et al. "Correct-N-Contrast: a Contrastive Approach for Improving Robustness to Spurious Correlations."
>
> [6] Kirichenko et al. "Last Layer Re-Training is Sufficient for Robustness to Spurious Correlations."
>
> [7] Haghtalab et al. "On-demand sampling: Learning optimally from multiple distributions."
>
> We hope that you can re-evaluate our work based on the clarifications above. If any concerns have remained unaddressed, we appreciate it if you let us know so that we can provide further details.

---

> > ### Comment · Reviewer_vLux · 2023-08-20
> >
> > I have read the replies by the authors and other reviewers. I appreciate the detailed explanations and references. While I feel a bit surprised by the general positivity from other reviewers, I do respect the consensus here and so I increased the score by 1.

---

> > > ### Author Response · Authors · 2023-08-21
> > >
> > > Thank you for your response and raising your evaluation. We would like to ask if there are still questions or concerns that lead to borderline rejection and we would be happy to discuss and address them as soon as possible.

---

### Author Rebuttal · Authors · 2023-08-10

We sincerely thank the reviewers for the feedback and especially the detailed suggestions that help us improve our manuscript. We appreciate that many reviewers commend our paper for its clarity, the alignment of our theoretical insights with empirical findings, the simplicity and novelty of our method, and the effectiveness of our method shown in experiments. In response to the questions raised, we've addressed each review with itemized answers. The updated additional tables and figures are organized into the 1-page PDF that we attach here. We hope this provides clarity and resolves any potential misunderstanding. Should there be additional comments or questions, we are more than happy to discuss them.

---

### Decision · Program_Chairs · 2023-09-21

**Decision:**

Accept (poster)

**Comment:**

This paper proposes a novel strategy to reduce the impact of spurious correlations
in the context of imbalanced datasets, and shows improvement both in practice
and in theory (under a simpler model). The reviewers appreciate the contribution
and clarity of the exposition, and after a thorough discussion seem to have arrived at a consensus to accept the paper.